



# The stable isotope composition of water vapour above Corsica during the HyMeX SOP1: insight into vertical mixing processes from lower-tropospheric survey flights

Harald Sodemann[1,2,3], Franziska Aemisegger[3], Stephan Pfahl[3], Mark Bitter[4], Ulrich Corsmeier[5], Thomas Feuerle[4], Pascal Graf[3], Rolf Hankers[4], Gregor Hsiao[6], Helmut Schulz[4], Andreas Wieser[5], and Heini Wernli[3]

[1]Geophysical Institute, University of Bergen, Norway
[2]Bjerknes Centre for Climate Research, Bergen, Norway
[3]Institute for Atmospheric and Climate Science, ETH Zürich, Switzerland
[4]Institute for Flight Guidance, Technical University Braunschweig, Germany
[5]Institute of Meteorology and Climate Research (IMK-TRO), Karlsruhe Institute of Technology (KIT), Germany
[6]Picarro Inc (now at Freeslate Inc., Sunnyvale, California, USA), California, USA

*Correspondence to:* Harald Sodemann (harald.sodemann@uib.no)

**Abstract.** Stable water isotopes are powerful indicators of meteorological processes on a broad range of scales, reflecting evaporation, condensation, and airmass mixing processes. With the recent advent of fast laser-based spectroscopic methods it has become possible to measure the stable isotopic composition of atmospheric water vapour in situ at high temporal resolution, enabling to tremendously extend the measurement data base in space and time. Here we present the first set of airborne spectro-

5 scopic stable water isotopes measurements over the western Mediterranean. Measurements have been acquired by a customised Picarro L2130-i cavity-ring down spectrometer deployed onboard of the Dornier 128 D-IBUF aircraft together with a meteorological flux measurement package during the HyMeX SOP1 field campaign in Corsica, France during September and October 2012. Taking into account memory effects of the air inlet pipe, the typical time resolution of the measurements was about 15–30 s, resulting in an average horizontal resolution of about 1–2 km. Cross-calibration of the water vapour measurements from

10 all humidity sensors showed good agreement in most flight conditions but the most turbulent ones. In total 21 successful stable isotope flights with 59 flight hours have been performed. Our data provide quasi-climatological autumn average conditions of the stable isotope parameters $\delta D$, $\delta^{18}O$ and d-excess during the study period. A time-averaged perspective of the vertical stable isotope composition reveals for the first time the mean vertical structure of stable water isotopes over the Mediterranean at high resolution. A d-excess minimum in the overall average profile is reached in the region of the boundary layer top due

15 to precipitation evaporation, bracketed by higher d-excess values near the surface due to non-equilibrium fractionation and above the boundary layer due to the non-linearity of the d-excess definition. Repeated flights along the same pattern reveals pronounced day-to-day variability due to changes in the large-scale circulation. During a period marked by a strong inversion at the top of the marine boundary layer, vertical gradients in stable isotopes reached up to $25.4‰\,100\,\mathrm{m}^{-1}$ for $\delta D$.





# 1 Introduction

The vertical distribution of water vapour in the atmosphere is highly variable. It is shaped by source, transport and sink processes, such as evaporation and condensation, horizontal and vertical advection, and mixing as determined by the atmospheric stability. The water vapour concentration itself does not provide information on the role different processes may play for its

variability. Considering in addition the stable isotope composition of atmospheric waters, termed here stable water isotopes (SWIs), can support the interpretation of the phase change history of water vapour in an airmass. SWIs are quantified by ratios between the concentration of the heavy stable isotopologues $H_2^{18}O$ and HDO and the most common isotopologue, $H_2^{16}O$, reported here in $\delta$-notation as $\delta D$ and $\delta^{18}O$ in the water vapour. The temperature at which these phase changes occur is the main driver of isotope fractionation which changes $\delta D$ and $\delta^{18}O$ (Dansgaard, 1964). SWIs are tightly coupled to atmospheric

transport as well as source (evaporation) and sink (condensation) processes (Jouzel et al., 1997), and can be seen as an integrating tracer of the water's Lagrangian transport history (Sodemann et al., 2008b). In the past, SWIs have been mostly used in a palaeoclimate context to infer past temperatures and moisture sources from natural archives (e.g., Jouzel et al., 1997) and in studies investigating the stratospheric water budget (e.g., Webster and Heymsfield, 2003). The process-based insight provided by SWIs, however, also extends to synoptic and sub-diurnal time scales, and to the lower troposphere, where most

atmospheric water vapour resides. In light of the key role of atmospheric moisture for the uncertainties in weather forecasts and climate predictions, understanding SWIs at these time scales promises to serve as an additional constraint for future model improvements.

Historically, the SWI composition of atmospheric water vapour has mainly been observed at the surface, or inferred indirectly from precipitation samples (e.g., Araguàs-Araguàs et al., 2000). The vertical distribution of SWIs in the atmosphere has been

sampled only on very few occasions due to the difficult sampling procedures. Ehhalt and Östlund (1970) were the first to report of $\delta D$ on airborne samples made inside Hurricane Faith in 1966. Their measurements of water vapour and ice crystals collected in several vertical layers allowed developing a hypothesis on the horizontal separation of updraft and downdraft shafts in Tropical Storms. Shortly thereafter, Taylor (1972) published a key dataset of the SWI composition in atmospheric water vapour from 20 flights made during 1967/68 over southern Germany. These SWI profiles showed substantial vertical

and temporal (daily to seasonal) variation. Atmospheric layers with remarkably different SWI composition were interpreted as resulting from the combined influence of turbulent mixing within the boundary layer, as well as advection and subsidence above an inversion-capped boundary layer. Rozanski and Sonntag (1982) used a Rayleigh-type condensation model to interpret an average profile of Taylor's (1972) data. This approach was disputed by Taylor (1984), noting that a model fit to averaged profiles will be meaningless unless the variability of each individual weather situation is taken into account. That controversy

emphasised both the importance of the meteorological conditions during individual flights, as well as the potential explanatory power of reliable climatological mean SWI profiles Rozanski and Sonntag (1984). Gedzelman (1988) devised a sophisticated multi-level adiabatic trajectory model to interpret the Taylor (1972) data by taking into account the pseudoadiabatic ascent and dry adiabatic descent of the measured airmasses, his study provided important insight into the control of cloud processing and vertical motion on the observed SWI profiles.



A second major SWI dataset was obtained by Ehhalt (1974); Ehhalt et al. (2005) from repeated sampling flights during 1966/67 and 1971-1973 over different parts of the US. The averaged profiles, obtained during different seasons and reaching up to 13 km altitude, revealed a general decrease of SWIs with height throughout the troposphere. Both season and location were significant for the variability of the mean profiles. Data points in and above the upper troposphere/lower stratosphere (UT/LS) region were less depleted than in the troposphere below, which was confirmed from balloon-based SWI sampling up to 20 km altitude (Pollock et al., 1980). Much of the further research was then focused on the UT/LS region, such as the collection of ice crystals in mid-latitude storm tops (Smith, 1992), and the airborne European Arctic transects at 400 hPa to investigate the moisture budget of the stratosphere (Zahn et al., 1998). The SWI distribution in the atmospheric boundary layer and lower troposphere have received less attention from airborne measurement campaigns. 20 years after the Taylor and Ehhalt flights, He and Smith (1999) obtained vertical profiles from 3 flights in the boundary layer and the lower troposphere. Albeit limited to few data points, their results allowed deriving a first estimate of the stable isotope flux from the surface to the atmosphere. A similar analysis was done by Tsujimura et al. (2007) for a number of flights in the boundary layer and lower troposphere over Mongolia.

All of the above flights suffered from the low temporal and spatial resolution available from a discrete sampling procedure. Samples were collected in cold traps, yielding only few samples per flight with low temporal and spatial resolution. SWI concentration was then determined after flight from the collected water vapour with isotope ratio mass spectrometry in a laboratory. Airborne in-situ SWI measurements have only become possible with the advent of laser spectrometry. Since the first airborne deployment of a laser spectrometer by Webster and Heymsfield (2003) in the tropical UT/LS, a number of studies have investigated SWIs in stratospheric water vapour and the tropical tropopause layer (Hanisco et al., 2007; Iannone et al., 2009; Sayres et al., 2009; Dyroff et al., 2010). In the context of validating SWI remote sensing products, Dyroff's instrument has recently been deployed in the troposphere during 7 flights in 2013 in the vicinity of the Canary Islands. The $\delta$D profiles acquired during these flights reveal very sharp vertical gradients related to airmass origin all the way through the troposphere (Dyroff et al., 2015). Similar features were apparent in $\delta$D measured during several airborne profiles up to $\sim$4500 m a.s.l. with a commercial laser spectrometer over interior Alaska (Herman et al., 2014). However, the SWI composition in the lower troposphere and boundary layer themselves have so far not been interpreted with respect the unprecedented information it can provide into the atmospheric transport history of water vapour.

The deuterium excess (d-excess$= \delta$D$-8 \cdot \delta^{18}$O), a second-order SWI parameter, is a widely used measure of the degree of non-equilibrium conditions during stable isotope fractionation. The parameter measures deviations of the SWI composition from the empirical global meteoric water line (GMWL) which relates $\delta^{18}$O and $\delta$D in global precipitation to one another with a ratio of 1:8 (Dansgaard, 1964; Gat, 2000). During phase-changes under non-equilibrium conditions, for example evaporation in an unsaturated environment that does not have time to reach equilibrium, the different advection speeds of the HDO and $H_2^{18}$O molecules lead to deviations from this empirical relationships. In that example, there would be relatively more of the light and fast HDO molecules in the vapor phase than one would expect from the GMWL, hence the name deuterium excess. On a global average, the precipitation has a d-excess of about 10‰, indicating that evaporation and precipitation generally take place under non-equilibrium conditions. Under strong evaporation conditions characterized by atmospheres with low relative





humidity, high d-excess, up to 40 have been observed in the vapor, in line with theoretical considerations (Merlivat and Jouzel, 1979; Pfahl and Sodemann, 2014), for example in the Mediterranean (Gat et al., 2003) or close to the sea-ice edge (Kurita, 2011).

Because of these properties, the d-excess can serve as a tracer of moisture origin and source conditions if it remains unaffected by the condensation history (Merlivat and Jouzel, 1979; Aemisegger et al., 2014; Pfahl and Sodemann, 2014). Moisture over the Mediterranean Sea is known to have relatively high values of d-excess. During a cruise in January 1995, Gat et al. (2003) acquired samples with a d-excess ranging from 10-34‰, which they related to intense evaporation along the coast into dry continental air. Based on measurements of d-excess at Rehovot, Israel, Pfahl and Wernli (2008) modeled the range of d-excess over Mediterranean moisture sources as 15–40‰ for individual cases. Taylor (1972) concluded from his data that the d-excess would not provide any additional value. He and Smith (1999) reported both $\delta$D and $\delta^{18}$O, but did not interpret the d-excess in their data. From airborne sampling of hydrometeors in Hurricane Olivia, Lawrence et al. (2002) found a pronounced day-to-day change in the d-excess, which was hypothesized to be due to evaporation condition changes. Current in-situ instrumentation is in principle capable of measuring all SWI species, yet this has not been exploited for interpreting the d-excess of lower tropospheric water vapour.

In this study, we report on airborne measurements of $\delta$D, $\delta^{18}$O and the d-excess obtained from a custom-modified commercial cavity-ring down spectrometer (CRDS) with an enhanced laser setup. Measurements were performed during a 30-day period in the boundary layer and the lower troposphere around Corsica, France, in the Western Mediterranean, yielding 21 successful flights with about 59 flight hours. The aircraft was based at Solenzara, Corsica, during the first HyMeX (Hydrological cycle in the Mediterranean experiment, Drobinski et al., 2014) special observations period (SOP1) in September/October 2012, providing many additional observations from different platforms (Ducrocq et al., 2014).

After describing the measurement campaign, data acquisition system and data processing (Sec. 2), we present and interpret a quasi-climatological autumn vertical SWI profile over the Western Mediterranean provided from all available SWI flight data (Sec. 3 and 4). Then, we present the temporal evolution of the SWI profiles during several days along a repeated flight track, exemplifying the tight coupling of SWIs to synoptic processes (Sec. 5). For a single flight, detailed back-trajectory analysis shows that the SWI composition, including d-excess, provides additional insight into moisture origin and evaporation processes (Sec. 6 and 7). Finally, we provide a summary and conclude with the main results of the study (Sec. 8).

## 2  Data

### 2.1  Airborne measurements during the HyMeX campaign

In autumn 2012 (5 Sep - 6 Nov 2012) HyMeX SOP1 took place in the western Mediterranean. In addition to the implementation of extensive ground-based field measurements on Corsica (Kalthoff et al., 2013; Barthlott et al., 2014), airborne measurements were carried out in the vicinity and above the island from 11 Sep to 11 Oct 2012, using the Dornier 128-6 (D-IBUF) research aircraft of the Institute of Flight Guidance, TU Braunschweig (Corsmeier et al., 2001). The dual-engine propeller aircraft has a typical cruising speed of $65\,\mathrm{m\,s^{-1}}$ and can reach altitudes of up to $24500\,\mathrm{ft}$ ($\sim$7300 m) using oxygen





masks in the non-pressurised cabin. The aircraft instrumentation provided a unique setup to simultaneously gather information on evaporation fluxes and the corresponding SWI composition in the marine boundary layer. Onboard instrumentation consisted of a comprehensive package for airborne meteorological flux measurements in the nose boom, acquiring humidity, wind vectors, temperature, and pressure at a 100 Hz sampling rate, as well as short-wave and long-wave radiation sensors for

the upper and lower half-space, radiative surface temperature and additional humidity sensors with lower sampling frequency (Table 1). The stable water isotope composition of ambient water vapour was measured using a commercial CRDS instrument (L2130-i, Picarro Inc.). The instrument was customised, resulting in a data acquisition rate of up to 2 Hz by a triple-laser setup allowing faster measurement of the different wavelengths for HDO and $H_2^{18}O$ molecules. During the HyMeX campaign, the instrument was used in a scheme with two ring-downs per data point, resulting in a 1 Hz data aquisition rate. Outside

air was guided into the system through a backward-facing cabin-mounted stainless-steel connected to a flushing pump (see Appendix A1).

In total 32 flights were performed (Table 2). During 11 flights (flights 17 to 27) the inlet pump TF2 was not working, leading to a mixture of cabin and outside air being sampled by the laser spectrometer. Here, only the 21 flights are reported during which all pumps were working. Figure 1 summarises the flight patterns for these 21 flights. The flight patterns were primarily designed

to provide information of the thermodynamic environment above and around Corsica leading to deep convection. The main aim was to study the processes related to moist convection, including surface evaporation and boundary-layer processes, over the island and the surrounding ocean. Thereby, flight patterns did both provide statistically robust sampling of the atmosphere at levels of constant altitude in the boundary layer and free troposphere, and profiles of the vertical structure of the atmosphere above land and water. Flight operations were focused on altitudes below 2000 m a.s.l. but reached above 3500 m a.s.l. for a few

vertical profiles. The flight patterns were often repeated after several hours or on the next day to investigate the changes in the atmosphere with the diurnal cycle or during a specific weather situation (Table 2).

## 2.2 Correction and calibration of humidity measurements

Reliable measurements of humidity are an important basis for the interpretation of the SWI data. Four independent humidity measurements were available on the aircraft: a dew point mirror hygrometer (TP), a humicap capacitive sensor hygrometer

(HC), a Lyman-$\alpha$ hygrometer (LY), and the CRDS $H_2^{16}O$ concentration (Table 1). The Lyman-$\alpha$ measures at very high frequency (100 Hz) but needs to be combined with the slower accurate measurements from either the humicap or the dew point mirror for bias correction. The combination of the Lyman-$\alpha$ with the dew point mirror (TPLY) provides a fast response to changing ambient conditions and generally stable measurements, even if sometimes after encountering very dry conditions it delivered bad data. In this study, the combined Lyman-$\alpha$/humicap humidity product (HCLY) was used as reference for calibrat-

ing the CRDS humidity measurements because it had a response time similar to the CRDS measurements. The correspondence between the CRDS and the HCLY humidity allowed to compensate for time shifts of the stable isotope measurements due to the inlet, piping, position in the aircraft and computer clock differences. Humidity measurements from the CRDS were then calibrated with the HCLY humidity data using a linear fit determined from each individual flight at a 1 Hz time resolution. The





slope and offset of this linear fit were quite stable between flights, but changed after the installation of a replacement pump (see Appendix A2).

An example for the uncalibrated 1 Hz humidity data from flight 1 is shown in Fig. 2a. Flights with strong turbulence showed slightly lower correlations. From the generally high correlation between the two humidity measurements at 1 Hz it is apparent that for $H_2^{16}O$ only very limited memory effects were introduced by the inlet system. During many flight periods the CRDS humidity measurement was more similar to the HCLY than to the humidity derived from the dew-point mirror/Lyman-$\alpha$ combination, as exemplified in Fig. 2b. Both aspects support that the CRDS instrument provided reliable measurements during most flight conditions, and was generally able to compensate for ambient pressure and temperature changes during flight (see Appendix A5).

## 2.3 Water vapour dependency correction of the SWI measurements

The CRDS measurements of the SWI composition of water vapour are affected by a mixing ratio dependent bias, and therefore require a water vapour dependency correction, sometimes also referred to as humidity-isotope response calibration, prior to or during calibration of the SWI themselves (Sturm and Knohl, 2010; Aemisegger et al., 2012; Bastrikov et al., 2014; Bailey et al., 2015). Humidity-isotope response calibration was performed in the laboratory in October 2012 (after the field deployment) using a WVISS unit (Los Gatos Inc.) for standard vapour generation and a stable isotope working standard also used for calibration of the measurements (Sec. 2.4). In autumn 2014 this was extended to lower humidities with the same CRDS and an improved setup using a dew point generator (Li-610, Licor Inc.). We measured the humidity-isotope response during stepwise increases of water concentration from 500 to 10000 ppmv, and fitted a power law function forced to zero at and above the reference level of 10000 ppmv for each isotopologue:

$$\Delta\delta^{18}O = a(q_{raw}^b - 10000^b), \tag{1}$$

$$\Delta\delta D = c(q_{raw}^d - 10000^d), \tag{2}$$

where the coefficients are $a = 1.0856 \times 10^4$; $b = -1.1068$; $c = 2.0578 \times 10^4$; and $d = -1.0774$. The calibration functions increase strongly for mixing ratios below 3000 ppmv (up to 2.7‰ for $\delta D$, 1.9‰ for $\delta^{18}O$). For d-excess, the correction reaches $-12.8‰$ at 3000 ppmv (Appendix A3), underlining the particular importance of humidity-isotope response calibration for this parameter. In the current generation of the L2130-i Picarro instrument the response is less pronounced than in earlier instrument generations (Aemisegger et al., 2012), but is different for each individual instrument (compare e.g., Bastrikov et al., 2014). We consider the humidity dependency of the SWI measurements as an instrument characteristic that remained constant during the campaign, as confirmed by repeated measurements during and after the campaign in 2012 (Appendix A3).

### 2.4 Stable isotope measurements calibration

Calibration of the CRDS instrument was routinely performed with the WVISS calibration unit using two working standards WS6 and WS9 (WS6; $\delta D$: $-78.68‰$; $\delta^{18}O$: $-10.99‰$; WS9; $\delta D$: $-166.74‰$; $\delta^{18}O$: $-70.19‰$), which approximately span





the range of the field measurements (albeit lower $\delta$D conditions were encountered). On every day with flight activity at least six calibration runs were performed with a duration of 10 min each. Before the flight, one calibration run at a water vapour mixing ratio of $\sim 20'000$ ppmv was done with both standards. After the flight two runs per standard at water vapour mixing ratios of $\sim 4'000$ and $\sim 20'000$ ppmv were performed. Ambient air dried with the WVISS unit was used as carrier gas for

the calibrations. Most of the calibration runs before the flights had to be discarded because it was later identified that the laser spectrometer and the WVISS had not yet sufficiently stabilised after start-up. Thus, with very few exceptions only the calibration runs carried out after the daily flight operations were used for calibration of the SWI measurements, using the constant part of each 10 min calibration period for averaging. As empirical stability criteria, the 1-$\sigma$ standard deviation during this period had to be below 3‰ for $\delta$D (2‰ for $\delta^{18}$O) for taking an individual calibration run into consideration. Calibrations

were then done following the procedure recommended by IAEA (2009).

No isotope calibrations were performed during flight due to lack of an on-board calibration system. In principle, instability can occur due to temperature and pressure fluctuations in the measurement cavity. Cavity pressure and temperatures inside the instrument are continuously stabilised, and records of these parameters indicate that no adjustment problems were encountered during most flight conditions (Appendix A5). Furthermore, the humidity measurement of the L2130-i is based on spectral

absorption of the $H_2O$ molecule and uses the same measurement principle as for the other isotopologues. Therefore, if the SWI measurements were affected by strong drifts during the flights we expect that this would also be detectable in the humidity measurements. This was not observed; the high correlation between the HCLY and CRDS humidity (Fig. 2) is therefore also considered as a strong indication for the quality of the SWI measurements obtained during flight. At least for our study with a relatively short flight time, regular calibration during flight would not necessarily improve the data quality, but substantially

reduce the available measurement time. It therefore seems that precise temperature and pressure regulation of the cavity and the other instrument parts are the most important requirements to limit the uncertainty of our airborne SWI data. Similarly, the study Dyroff et al. (2015) indicated that for their instrument exact temperature control of the instrument components rather than calibration during flight was the main factor required for obtaining data at sufficient quality.

## 2.5   Data quality and resolution

Memory effects due to interaction with the piping walls are larger and species dependent for $H_2^{18}O$ and HDO compared to $H_2^{16}O$. From laboratory experiments with the inlet system used in the aircraft, a time constant of 2.3 s for $\delta$D (1.7 s for $\delta^{18}$O) was determined for switches from low to high isotope concentrations and vice versa at constant $H_2^{16}O$ concentrations, compared to 1.3 s for changing $H_2^{16}O$ concentrations (Aemisegger, 2013). This causes smoothing over $\sim$2–3 data points at 1 Hz time resolution.

Stability tests of the CDRS SWI measurements were performed before, during and after the HyMeX campaign using the WVISS as a standard vapour source (Aemisegger, 2013). For these tests, calibration standard WS6 was measured for several hours at a water vapour mixing ratio of $\sim$9000 ppmv. Stability was assessed using the Allan deviation. The stability of the measurements degraded during the campaign, possibly because of modifications to the mounting of the optical fibers inside the CRDS, which had to be done for the flight certification of the instrument. These issues also affected instrument stabilisation





and increased warm-up time (up to 2 h). While any resulting drifts were removed by calibration, precision generally decreased. At the averaging times considered here this mainly affected the d-excess during the later flights of the campaign.

From long-term stability tests, the standard deviation of the measurements during the campaign for an averaging time of 60 s was 2.3‰ for $\delta D$, 0.36‰ for $\delta^{18}O$ and 1.17‰ for d-excess (Table 3). The dependency between averaging times, horizontal and vertical resolution, and standard deviation for all isotopic parameters is summarized in Table 3. The 1-$\sigma$ standard deviation increased considerably at lower water concentrations (Appendix A4. Generally, the practical meaning of the standard deviations is somewhat limited, as the atmospheric SWI composition exhibited substantial variability on time scales of seconds to minutes, in particular in the free troposphere (see below). As changing evaporation conditions are expected to create signals in the d-excess on the order of 10‰ and more, significant differences are detectable at a 30–60 s averaging time, providing d-excess with a spatial resolution of up to 2–4 km horizontally and 150–300 m vertically when ascending at 5 m s$^{-1}$ (Table 3). For the analysis presented here, we report 15 s averages for $\delta^{18}O$ and $\delta D$, and averages for the d-excess unless stated otherwise.

## 2.6 Meteorological background conditions

The climate of Corsica in autumn is characterised by warm sea water conditions and progressively increasing influence of cold airmasses from northern directions. Climatological rainfall totals in Bastia (42°33'N; 9°29'E) increase from 65 mm in September to 110 mm in October (the wettest month of the year), albeit with large inter-annual variability. Isolated deep convection frequently develops over the mountainous island in autumn favoured by an inland transport of water vapour via thermally driven circulations (Barthlott et al., 2014; Adler et al., 2015). At the east coast north of Solenzara, 2 m air temperatures measured at the energy balance station San Giuliano ranged between about 20–27°C during the day and 15–20°C at night during the campaign period, with a rainfall total of 222 mm for the two-month period from 25 Aug 2016 to 25 Oct 2016.

The climatological SWI composition of atmospheric moisture and sea water in this region is not known. Non-representative river runoff samples taken at two elevations in the eastern part of the island during the campaign suggest weakly depleted precipitation and enhanced non-equilibrium fractionation at the moisture source prior atmospheric transport ($-6.9 \pm 0.2$‰ in $\delta^{18}O$ and $13.9 \pm 1.6$‰ in d-excess at 1000 m a.s.l.). The signal was similar but weaker further downstream, possibly due to evaporation of ground water and river runoff ($-6.2 \pm 0.2$‰ in $\delta^{18}O$ and $9.7 \pm 1.6$‰ in d-excess at 15 m a.s.l.). Coastal sea water from Solenzara in contrast was enriched, and showed indications of non-equilibrium fractionation due to ocean evaporation ($1.7 \pm 0.2$‰ in $\delta^{18}O$, $-4.0 \pm 1.8$‰ in d-excess from two samples). Due to the small sample size these values only provide indicate relative differences.

Vertical profiles of average humidity, temperature and wind speed from all flights with valid SWI measurements (Table 2) are shown in Figure 3. Humidity decreases fairly linearly with height, and has largest variations at low levels, ranging between 5–17 g kg$^{-1}$ (Fig. 3a). Above 1500 m a.s.l., very low humidity conditions were encountered, while variability remained substantial up to the highest flight altitudes. Temperatures ranged between 18–32°C at the ground, and decreased with an average lapse rate of 5.1 K km$^{-1}$ (Fig. 3b). Between elevations of 1500–3000 m a.s.l., some excursions to colder than average conditions can be noted, which were associated with advection of cold air from northerly directions during flight 4 and 5 (not shown). Horizontal wind speeds were generally weak during the flights, mostly below 10 m s$^{-1}$ near the ground, with some excursions of above





20 m s$^{-1}$ (Fig. 3c). Average wind was increased above 2000 m a.s.l. in the free troposphere, but remained at a moderate 10 to 15 m s$^{-1}$. Thus, the typical advection speed was below 400 km day$^{-1}$ in the boundary layer, and on the order of 800–1200 km day$^{-1}$ in the free troposphere.

The aircraft operated outside of clouds (visible flight rules, VFR) throughout the campaign. Therefore, relative humidity with respect to liquid (RH) remained below saturation at the aircraft location for $< 99\%$ of all data points (not shown). Average cloudiness during the campaign was also low, as subsidence dominated on the large scale. The exception were several days with strong thunderstorms at the beginning of the campaign period (02 to 05 Sep 2012).

## 3   Mean vertical SWI profiles

Figure 4 displays the averaged SWI composition in the Western Mediterranean from all corrected and calibrated data from the 21 flights. The measurements reveal an unprecedented level of detail on the SWI distribution in the Western Mediterranean lower troposphere. The 200 m binned vertical profiles of the main isotopes $\delta$D and $\delta^{18}$O have distinct, coherent shapes, showing an overall decrease from close to the surface ( $\delta$D = $-96.7$‰ and $\delta^{18}$O = $-14.1$‰ between 0 and 200 m a.s.l.) to the free troposphere ( $\delta$D = $-225.1$‰ and $\delta^{18}$O = $-30.2$‰ between 3400 and 3600 m a.s.l.; Fig. 4a and b, thick red lines). The vertical gradients were -36.7‰ km$^{-1}$ for $\delta$D and $-4.2$‰ km$^{-1}$ for $\delta^{18}$O, respectively. At all elevations, the distribution is skewed towards the minimum values, as is apparent from the horizontal stripes of data points acquired during longer horizontal transects. For this reason, the bin minimum and maximum values are discussed, rather than the standard deviation. The bin maximum values decrease less strongly than the bin means with vertical gradients of $-16.7$‰ km$^{-1}$ for $\delta$D and $-2.1$‰ km$^{-1}$ for $\delta^{18}$O (Fig. 4a and b, red dashed lines). The bin minimum values show a distinct decrease between 1700 and 2300 m a.s.l., which is to some extent reflected in the mean and also leads to stronger vertical gradients of the minimum values ($-51.9$‰ km$^{-1}$ for $\delta$D, $-6.0$‰ km$^{-1}$ for $\delta^{18}$O). While there are indications for further decreasing mean values above 3700 m a.s.l., data are only available from one flight and are thus not sufficient to extend the bin averaging to these elevations.

The shape of the composite vertical profile of the d-excess is different from that of the primary stable isotope parameters (Fig. 4c). Mean values show a maximum close to the surface (16.0‰ between 0 and 200 m a.s.l.), decrease to an overall minimum bin mean of 4.9‰ between 1200 and 1400 m a.s.l., and then increase again to bin mean values of 8 to 9‰ above 2800 m a.s.l. The bin minimum values are negative throughout the profile (overall minimum $-15.7$‰ at 1300 m a.s.l.). The d-excess maximum values show a marked linear decrease from 60.2‰ close to the surface (between 0 and 200 m a.s.l.) to about 21.2‰ at 1500 m a.s.l., and again a marked increase above 1700 m a.s.l. to relatively uniform maximum values around 60‰ above 2000 m a.s.l. Many of the high d-excess values aloft have been sampled during relatively dry conditions ($q < 2$ g kg$^{-1}$, indicated by the blue dots). The overall minimum in d-excess at 1300 m a.s.l. is an interesting finding, that is further investigated in Section 4.2.

It is known from several other studies that upper-tropospheric air can have a high d-excess value (Galewsky et al., 2011; Samuels-Crow et al., 2014). They are partly due to the definition of the d-excess as a function of $\delta$D and $\delta^{18}$O with a constant slope and to the non-linearity of the delta scale. The high d-excess encountered above the marine boundary layer is therefore not





an indication of insufficient data quality, but a real feature of the SWI composition in the atmosphere. That said, the precision of the d-excess measurements from the CRDS instrument (in terms of signal-to-noise ratio) decreased substantially at very low humidity during flight (see Appendix A4).

### 3.1 Comparison to previous aircraft measurements of stable water isotopes

We now compare our Western Mediterranean measurements to previous airborne SWI measurements in other regions to investigate the similarity to our data. Only few studies reported detailed profiles of SWIs in the lower troposphere from a range of different atmospheric conditions. A comprehensive sampling of SWIs in tropospheric water vapour was carried out by Ehhalt (1974) during 1965–67 and 1971–1973 over three marine and continental sites in the US. These $\delta$D data are from water vapour collected in cold traps during flight legs at constant altitude for 10–60 min. The data cover a large variety of weather con-

ditions from all seasons and different climatic regimes (Fig. 5a). The correspondence between the envelope of the HyMeX data (Fig. 5a, black lines) and the Ehhalt data is remarkable for the samples above Scottsbluff, Nebraska (Fig. 5a, red dots, no data below 1500 m a.s.l.). The HyMeX data are more depleted above 2000 m a.s.l. than their measurements over the Pacific Ocean, but agree well at lower altitudes (Fig. 5a, blue circles). In comparison to Death Valley, California, the HyMeX data were generally more enriched near the surface and showed lesser gradient with height, indicating well-mixed conditions above

the desert (Fig. 5a, green crosses).

He and Smith (1999) reported the SWI composition of trap-collected water vapour from three flights over the forests of New England (Fig. 5b and c). For the $\delta$D, there is generally good correspondence between their data points from the two summer flights (Fig. 5b, blue and green lines) and the average HyMeX profile (thick black line). The d-excess data fit within the envelope of the HyMeX data (Fig. 5c). Note however that because of the low humidity, He and Smith (1999) estimated

the upper part of the red profile from all other measurements during their flights. Overall, the HyMeX data are thus in good correspondence with previously acquired tropospheric aircraft data, both in terms of the range and vertical variability, while adding a large amount of information on the shorter time scales and smaller spatial scales. In the next section, we attempt an interpretation of the HyMeX SWI data in terms of processes shaping the observed vertical structures.

### 4 Processes shaping the observed mean profiles

We now explore two hypotheses on how the observed profiles could have been caused. The composite profiles (Fig. 4) likely reflect the substantially different SWI composition of water vapour in the (mostly marine) boundary layer and the depleted free troposphere above. A common interpretation of the vertical gradient of the SWI is the increasing depletion due to a Rayleigh fractionation process (Gedzelman, 1988; Rozanski and Sonntag, 1982). Thereby, fractionation during a moist adiabatic ascent leads to progressive depletion higher up in the atmosphere, as can be described in terms of a simple model. The mean values in

the free troposphere (blue values above 1700 m a.s.l.) decrease in line with what can be expected from Rayleigh fractionation (see below). Alternatively, adopting an airmass mixing perspective, the maximum values of the primary isotope parameters could reflect the end member of water vapour from the sea surface, close to equilibrium conditions, which due to typically





well-mixed conditions is constant up to almost 1200 m a.s.l. The slowly decreasing maximum values above are then a result of the (increasingly less likely) detrainment of such end-member boundary-layer air to the free troposphere and vice versa. The minimum values of the major isotope species above 1700 m a.s.l. with their very depleted conditions would then reflect airmasses of different, higher-elevation origin, for instance due to descending airmasses. Then, the minimum values below

1700 m a.s.l. might reflect entrainment and mixing of such depleted water vapour into the boundary layer. As descending airmasses are typically dry, the impact on the isotope composition will be lower during downward mixing than upward mixing. This is in agreement with the larger vertical gradient of the minimum values in the boundary layer compared to the maximum values. The plausibility of these two hypotheses is further investigated in the following sections.

### 4.1   Specific humidity dependence of $\delta$D

One way to display the additional information contained in the SWIs is the $\delta$D-q diagram (e.g., Noone, 2012). The data cloud in the $\delta$D-q diagram using all valid HyMeX data (Fig. 6a) can be interpreted as the range of different Rayleigh fractionation regimes the moist air experienced (e.g., Rozanski and Sonntag, 1982; Worden et al., 2007). Initially, the boundary-layer air below 1700 m a.s.l. (blue and red data points) is quite moist ($\sim$10–20 g kg$^{-1}$) and ranges mostly between $-100$‰ to $-70$‰ for $\delta$D, in isotopic equilibrium with a source $\delta$D of 0‰ at SSTs between 8°C and 33°C, respectively, with an average of

21°C. This matches with typical SST observations around Corsica obtained from the IR surface thermometer (e.g. 23.0$\pm$4.7°C during flight 09). The increasing depletion of the data with decreasing specific humidity could then reflect a typical equilibrium fractionation pathway expected from a moist adiabatic condensation process (the canonic Rayleigh fractionation, e.g., Gat, 1996). The dashed lines in Figure 6a show some Rayleigh processes starting at different initial conditions. The Rayleigh process requires however saturated moist adiabatic ascent, without exchange with the surroundings, a condition hardly observed during

the mostly cloud-free HyMeX campaign period. Furthermore, the Rayleigh fractionation lines seem to follow a too steep depletion pathway compared to apparent lines in the data set that connect the more and less depleted data points in Fig. 6a.

An alternative interpretation of the $\delta$D-q diagram is to consider these apparent curves as mixing lines between two end members of different SWI composition (Noone, 2012). Then, one end member is the free troposphere air with low specific humidity and strong depletion, and the other is boundary layer air with high specific humidity and weak depletion. Different

blends due to vertical mixing processes then produce bent mixing lines in the $\delta$D-q diagram. Such mixing lines are shown by the solid curves in Fig. 6a. As an example, the detrainment of isolated plumes of moist air with a typical specific humidity of 12 g kg$^{-1}$, forced upward by ascent at the steep topography of Corsica, and subsequent mixing at a ratio of 1:6 with a dry and depleted end member (for example q = 2 g kg$^{-1}$; $\delta$D = $-250$‰) will provide an airmass with 3.4 g kg$^{-1}$ specific humidity and $-168$‰ $\delta$D.

Three arbitrary mixing scenarios are shown as solid colored curves in Fig. 6a that could explain a large share of the data. When a relatively moist and moderately depleted air mass (5.0 g kg$^{-1}$; -200‰) is repeatedly mixed with an equal volume of moister and less depleted boundary-layer air (16 g kg$^{-1}$; -90%), the solid black mixing line results. This mixing line is a lower-right (depleted $\delta$D and high humidity) bound to most of the blue and red data points, and represents shallow mixing within the lowermost 1700 m a.s.l. The solid magenta line has a boundary-layer end member with -80‰ $\delta$D and 15.0 g kg$^{-1}$





specific humidity, and a free-troposphere end member at $-300‰$ and $1.0\,\mathrm{g\,kg}^{-1}$. This mixing line provides an upper-right bound (enriched $\delta$D and high humidity) to the low-elevation data (blue and red data points), which would result from mixing with free-troposphere air.

The black data points measured above 1700 m a.s.l. generally cover drier conditions and stronger isotopic depletion. The solid green curve shows an exemplary mixing line resulting from a moist and less depleted end member in the upper boundary-layer ($8.0\,\mathrm{g\,kg}^{-1}$; $-100‰$) and a very dry and depleted free tropospheric air mass ($0.5\,\mathrm{g\,kg}^{-1}$; $-220‰$). While this curve explains some of the most enriched low humidity values, many other combinations are possible. In general, this analysis shows that a substantial part of the data can be explained by two kinds of mixing processes, one taking place between moisture originating from surface evaporation and the boundary layer, and one between moderately depleted and moist air from the upper boundary layer which is entrained into the dry free troposphere.

While our measurements were done almost exclusively in unsaturated conditions, fractionation during Rayleigh-type condensation processes is obviously required to obtain depleted end members in the first place. However, this depletion process during a moist adiabatic ascent can have taken place far away, for example in tropical deep convection or a warm-conveyor belt like ascent related to an extratropical cyclone, before being advected to the Mediterranean free troposphere (see Sec. 7).

## 4.2 Specific humidity dependence of the d-excess

A similar analysis can be done for the non-equilibrium fractionation indicator d-excess. The cloud of all valid HyMeX measurement data can be separated into three regions with different characteristics (Fig. 6b). The d-excess measurements acquired between 500 to 1700 m a.s.l. are located in a regime of high specific humidity with a mean of about $6\,\mathrm{g\,kg}^{-1}$ (blue data points). In this range, d-excess averages at about 0-10‰, but with distinct excursions into much higher values as shown by the bin average (black line). The red point show d-excess measurements acquired below 500 m a.s.l. Here, d-excess ranges between 0 and 40‰. Many of the high d-excess values were acquired at low-elevation transects over the ocean, thus likely representing the imprint of the non-equilibrium fractionation conditions due to the low relative humidity with respect to SST during evaporation.

The branch of free-troposphere measurements (>1700 m a.s.l., Fig. 6b, black dots) extends from a d-excess of about 0‰ at $\sim5\,\mathrm{g\,kg}^{-1}$ to progressively higher values of up to 60‰ as the specific humidity decreases to below $0.5\,\mathrm{g\,kg}^{-1}$. As noted in Section 2, the d-excess data beyond this value are strongly affected by the correction for the humidity dependence of the CRDS SWI measurements. It is noteworthy that almost all measurements at low humidity have high d-excess values, and that there is an apparently seamless transition from high d-excess at below $2\,\mathrm{g\,kg}^{-1}$ to lower d-excess at $2$–$4\,\mathrm{g\,kg}^{-1}$. This argues against a technical artifact as explanation for the very high d-excess, and for an actual geophysical signal.

The solid green line in Fig. 6a shows the d-excess expected from a Rayleigh fractionation process starting at $10°$C surface air temperature and an isotopic composition of $\delta$D$=-86‰$ and $\delta^{18}$O$=-12‰$ (d-excess$=10‰$). The shape of the curve illustrates that equilibrium fractionation can strongly impact the d-excess due to the different temperature dependency of the fractionation factors (Sodemann et al., 2008a, e.g., Duetsch et al., manuscript in preparation). Note that the displayed green line provides an





upper bound to the increase of d-excess with decreasing humidity. Mixing processes generally seem to act towards lowering the d-excess relative to that upper bound.

The combination of the high d-excess in the free troposphere, and the data points influenced by surface evaporation produce an apparent minimum of the d-excess at ~4 g kg$^{-1}$, in correspondence with the vertical profile (Fig. 4c). In addition, there are many data points with d-excess below 0‰, mostly at specific humidity between 2–8 g kg$^{-1}$. Negative d-excess in water vapour can be due to evaporation of rainfall or from continental moisture sources due to strong soil evaporation. During evaporation, the d-excess of raindrops is reduced (Aemisegger et al., 2015), along with positive d-excess values in vapour, before ultimately a negative imprint is left behind when the drops evaporate completely (e.g., Gat, 1996). Evaporating rainfall or dissolving (evaporating) clouds could thus create a vertical gradient of d-excess in vapour under certain conditions. Thereby, vapour with decreasing d-excess is contributed to the environment as more of the raindrops evaporates during their fall. We speculate that near the boundary-layer top the contribution of rainfall and cloud evaporation to the vapour is largest. One could for example conceive a situation where boundary-layer top cumuli locally produce rainfall in a saturated environment, whereas evaporation occurs in the immediate vicinity and below the cloud in regions influenced by dry-air entrainment from the free troposphere above. Being relatively far away from the high d-excess created during surface evaporation, this combination of processes could potentially shape an overall d-excess minimum in the vertical profiles. The predominance of d-excess values close to 10‰ as in the global meteoric water line, rather than the occasional high values representing enhanced non-equilibrium fractionation during evaporation suggests that overall evaporation conditions at the moisture sources were not characterised by strong non-equilibrium conditions, or that newly evaporated water contributed comparatively little to the available moisture in the boundary layer. In our data set, the high d-excess signal from local, short-lived intense evaporation events thus seems to disperse with altitude. Benetti et al. (2014, 2015) highlight the importance of mixing processes between surface vapour and the free troposphere in the marine boundary layer. From analysing ship-based SWI measurements in the subtropical East Atlantic with an idealised model, Benetti et al. (2015) conclude that mixing with free troposphere air is more important for the primary isotope parameters than for d-excess, which is mainly controlled by non-equilibrium effects during evaporation, a finding that may be specific to evaporation-dominated situations. In this comparison, it is important to note that the lowest measurement altitude of the aircraft was at 150 m a.s.l., making direct comparison to ship-based measurements difficult. Gat et al. (2003) even noted a vertical SWI gradient for different measurement altitudes on a ship. Thus, the influence of the free troposphere at our lowest measurement altitude may already be higher than for ship-based measurements. This also points to deviations from the well-mixed conditions of SWI in the marine boundary layer assumed above.

## 5 Evolution of the vertical stable isotope structure during several days in sequence

The contrast between weakly depleted water vapour in the boundary layer and moderately to strongly depleted water vapour in the free troposphere is further explored now by considering the atmospheric environment and airborne measurements during specific flights. As will be exemplified below, diurnal variability and larger-scale processes jointly shape the vertical stratification reflected in the SWI profiles.





A sequence of three flights was executed following pattern #3 between 20-21 Sep 2012 (flights 09–11). The pattern consisted of several east-west transects across the island, a low-level leg over the sea to the west, and vertical profiles over the island and to the east and west of it (Fig. 8a, white line; Fig. 1, green line). The vertical profile northwest of Corsica was also included in pattern #1, which was flown after the sequence, on 23 Sep 2012 (flight 12). This sequence of flights allows for investigating

the development of the SWI vertical profiles for 4 flights and during 4 days. The period from 20 to 23 Sep 2012 was marked by pronounced differences in the vertical stratification of the lower and middle troposphere. On 20 Sep, ex-Hurricane Nadine was located at the upstream side of an upper-level ridge near 30°W and 35°N, in close proximity to an equatorward extending PV streamer (Fig. 7, thick black contour). Interaction between these two dynamical features determined the synoptic evolution over the western Mediterranean during the following days.

Initially, the eastern flank of a large upper-level ridge was located over Corsica, leading to the advection of airmasses from the northwest with wind speeds of $\sim$16 m s$^{-1}$ on 500 hPa (Fig. 8a, white arrows; Fig. 7, black contour). Central Europe was under the influence of a high pressure system, and the Mediterranean experienced intermediate temperatures, with 282–286 K on 850 hPa (Fig. 7, color shading). The region of interest was almost cloud-free (Fig. 8a, shading). Winds were from northerly directions with speeds of about 7 m s$^{-1}$ on 700 hPa and associated with cold-air advection throughout the lower troposphere

(Fig. 8a, red arrows). During the first flight of this sequence (flight 09) at 20 Sep 2012, a vertical profile was flown at the northwest end of flight pattern #3, west of Corsica over the Mediterranean Sea (Fig. 8a, cyan line).

Figure 9a-d displays the temperature, specific humidity, and SWIs at the vertical profile. The temperature profile depicts two pronounced inversions between 900-1000 m a.s.l. and 2300-2500 m a.s.l., which are well reproduced during ascent and descent (Fig. 9a, black and grey line). The boundary layer below 900 m a.s.l. and the layer above 2500 m a.s.l. are both well-

mixed with uniform lapse rates of 7.3 and 6.0 K km$^{-1}$, respectively. The intermediate layer has a lapse rate of -7.6 K km$^{-1}$, and an approximately isothermal lower end that shows enhanced temperature variability. The profile of specific humidity shows changes corresponding to the observed thermal structure (Fig. 9b). The boundary layer has a specific humidity of $\sim$9–11 g kg$^{-1}$, which drops to substantially lower values (3–5 g kg$^{-1}$) at and above the capping inversion. As for temperature, humidity variations are enhanced between 1000 m a.s.l. and 1700 m a.s.l., possibly related to the partial cloud cover observed

in the satellite image (relative humidity reached 90% between 600 and 900 m a.s.l.). Specific humidity drops to uniformly very low values (1–2 g kg$^{-1}$) at the upper inversion. Some difference between the upward and downward profile is apparent for humidity (Fig. 9b, black and grey line).

The vertical profile of $\delta$D (Fig. 9c) in turn corresponds to the humidity profile. In the boundary layer values of weak depletion (-90‰ to -100‰ $\delta$D) reflect proximity to the evaporation source. At the capping inversion (900 m a.s.l.) $\delta$D drops

to -150‰, again with increased variability. Above 1700 m a.s.l. $\delta$D increases again to -120‰. This isotopic inversion together with the specific humidity information confirms that the layer between 1000 and 2300 m actually consists of two layers of different origin or airmass history. Without the SWI measurements this distinction could hardly be made. Repeatability is lower for the upward/downward profiles for $\delta$D, in particular above 2300 m a.s.l. The downward profile (black line) is expected to be more reliable, as memory effects are smaller when moving from dry to moister conditions during descent. Note that

the coherent variability across the three available specific humidity measurements point to at least a partial contribution from





spatial variability (not shown). A further influence may come from the pressure dependency of the CRDS measurements during upward and downward motion of the aircraft (Appendix A5).

The shape of the d-excess profile (Fig. 9d) does not reflect the thermodynamic structure observed in temperature and humidity. In the lowermost 300 m a.s.l. values of up to 40‰ are measured for the d-excess. This likely reflects non-equilibrium

fractionation during intense evaporation in the marine surface layer at the aircraft location or upstream. As a proxy for the d-excess during evaporation, relative humidity with respect to SST ($RH_{SST}$) has been calculated from $q_{HCLY}$ at flight altitude and the skin temperature measured from the infrared radiometer (Table 1). The observed d-excess of ∼30‰ corresponds to an $RH_{SST}$ of ∼40% (Pfahl and Sodemann, 2014), slightly lower than the measurements along a low-level transect immediately following the profile ($RH_{SST}$ of ∼48% at 150 m a.s.l., Table 4b, second row). It is noteworthy that the SST is more than 5 K

warmer than the air temperature at 150 m a.s.l., which suggests that the at the elevation of the aircraft measurements the atmosphere may not yet be fully coupled to the surface conditions at this time of day. Nonetheless, the d-excess indicates strong non-equilibrium fractionation conditions at the underlying sea surface during intense evaporation. The d-excess decreases to ∼15‰ in the layer between 300 and 1500 m a.s.l., shows values near 10‰ between 1500 and 2300 m a.s.l., before increasing to more than 60‰ at the highest part of the profile. The d-excess is fairly well reproduced for the upward and downward profile,

with most pronounced differences in the very dry part of the profile above 2300 m a.s.l.

The same flight pattern was repeated 23 h later during flight 10 on 21 Sep 2012 (Fig. 8b). The large-scale circulation remained very similar to the previous flight (Fig. 7). Around Corsica, a weak southeasterly flow advected dryer air towards the island at lower-troposphere levels and reduced cloudiness (Fig. 8b). Upper-level winds had strengthened and turned to westerly directions. The vertical profile of temperature had changed considerably, now showing an inversion between 1700 and 2200 m

a.s.l. (Fig. 9e). The upper layer had warmed by about 3 K and dried uniformly to a specific humidity of 1-2 g kg$^{-1}$ (Fig. 9f). The vertical contrasts in the SWI composition had increased markedly. Below the inversion δD was slightly more depleted compared to 23 h before (−110 to −120‰, Fig. 9g). This is quite far from equilibrium with the (δD= −80‰ for an SST of ∼ 20°C), even though the sea surface had cooled by ∼6 K (Table 4b). Across the inversion, the δD decreased to −380‰ within only 300 m vertically during the downward profile (Fig. 9g, solid line). At the highest elevations (3700–3300 m a.s.l.)

the δD profile showed very good reproducibility of the upward and downward profiles with about −250‰ depletion. The d-excess profile had a similar shape as during the previous flight, albeit at lower elevation, and agreed very well for both profiles (Fig. 9h). A minimum d-excess of about 10‰ was measured just below the temperature inversion at ∼1700 m.

The flight pattern was repeated a third time another 3.5 h later during flight 11, 11–15 UTC 21 Sep 2012, with largely unchanged large-scale circulation and winds around Corsica (not shown). The temperature inversion had descended to 1400 m

a.s.l. (Fig. 9i). Contrasts in specific humidity were very pronounced (Fig. 9k), with a moister boundary layer than during flight 10 (∼9 g kg$^{-1}$, compare Fig. 9f), a very dry intermediate layer (1 g kg$^{-1}$) and a moderately moist upper layer (3–4 g kg$^{-1}$). The drastic humidity difference between the two lowermost layers was also apparent in the δD profile, with a jump from −100‰ to −200‰ within about 200 m vertical distance (Fig. 9k). Reproducibility was very good for this δD profile, except for the dry intermediate layer. The d-excess showed a quite different profile for this flight with values of up to 30‰ in the dry

layer (Fig. 9l). Possibly, low d-excess water vapour from the surrounding may have reduced the d-excess by entrainment. The





absence of a clear maximum near the bottom indicates that high RH conditions (70–80%) and weak evaporation prevailed in this situation (d-excess of ∼14‰).

Another 36 h later, at the final flight in this sequence (flight 12 on 09 UTC 23 Sep 2012), the PV streamer in the large-scale circulation had evolved into a cut-off with rapid cyclogenesis near the Gulf of Biscay, leading to the advection of air with high

equivalent potential temperature from the Sahara to the Western Mediterranean (not shown). Southwesterly flow dominated throughout the troposphere, associated with widespread broken cloud cover (Fig. 8c). The single downward vertical profile available on 23 Sep 2012 shows that the vertical structure of the boundary layer had changed significantly in all variables (red profiles in Fig. 9i-l). Temperature had increased by about 2-3 K, the layer above 1500 m a.s.l. had moistened significantly, leading to the observed cloudiness. The $\delta$D profile had shifted to uniformly less depleted conditions of around -140‰ above

1400 m a.s.l, while The d-excess gradually decreased from values around 20–30‰ below 500 m a.s.l. towards about 0‰ at the top of the profile.

The temporal variability of the vertical SWI structure observed above has implications for the relative importance of vertical mixing and large-scale advection for understanding the vertical profiles. It appears that the two processes go hand in hand: the inversions delineate layers that are internally well mixed but evidently have weak mixing across the inversion. Inversions

thus also separate layers with a very different origin and transport history. The combination of the two can lead to strongly variable and well-structured vertical profiles as observed during the HyMeX flights, which show surprisingly strong day-to-day variability also in a situation where the large-scale conditions are seemingly rather similar. One implication of this is that a single vertical profile in a high-pressure situation might not be representative for this weather situation. The presence of the complex topography around Corsica with various influences in upstream and downstream regions further complicates

the situation. The following sections exemplify for one of the flights discussed above (flight 10) how stable isotopes allow to interpret the transport history of an airmass in more detail.

## 6  High-resolution SWI measurements during flight 10 (21 Sep 2012)

We now present the high-resolution horizontal and vertical variability obtained during flight 10 at 21 Sep 2012 (the second flight discussed in Sec. 5). During this flight clouds were almost absent, and a strong vertical wind shear prevailed, with

southeasterly winds in the boundary layer and westerly winds in the free troposphere (Fig. 8b). A deep inversion of ∼3 K between 1700–2200 m a.s.l. characterised the atmospheric stratification during this flight, both at locations west (Fig. 9e) and east (not shown) of Corsica, separating a moist marine boundary layer from a very dry free troposphere (Fig. 9).

Figure 10 shows the flight altitude, meteorological and SWI parameters during the first 2.5 h of flight 10 (flown along pattern #3, compare Fig. 1). The flight period has been subdivided into segments A to H in Fig. 10. During the vertical profile in

segment A, the aircraft encountered dramatically different meteorological and SWI conditions between flight altitudes of 1500 to 2500 m a.s.l. At 0700 UTC, during ascent from about 1700 to 2000 m a.s.l., specific humidity decreased abruptly from 8 to about 1 g kg$^{-1}$ (Fig. 10c, red and black lines). Humidity was in fact below detection limit for the dew point mirror hygrometer. Associated with the drop in specific humidity were an increase in potential temperature from 299 K to 305 K (Fig. 10b, black



line), an increase in air temperature from 10 to 12°C (Fig.b, red line) and a drop in relative humidity (RH, Fig. 10d) from 80% to ∼15%. Winds gradually changed from southerly to northwesterly directions, and back, during segment A (Fig. 10e).

The stable isotope composition shows similar strong changes across the inversion at the boundary-layer top. $\delta D$ ($\delta^{18}O$) shows an initial drop from $-100‰$ to $-225‰$ ($-13‰$ to $-35‰$), and then a further depletion before the re-entry into the boundary layer at 0711 UTC with a minimum of $-275‰$ ($-41‰$ for $\delta^{18}O$) at the end of segment A (Fig. 10f and g). The d-excess mirrors the jump of the primary SWI parameters, increasing from 12‰ to 65‰ across the inversion at 0702 UTC (the d-excess is drawn in red for $q < 2\,g\,kg^{-1}$). It then decreases to about 40‰ for 5 min when the aircraft is above 3000 m a.s.l., before increasing again to about 55‰ (the period when the primary SWI parameters are lowest), before finally returning to about 15‰. The d-excess thus hints towards a different airmass above 3000 m a.s.l. at this location, which is supported by close inspection of the wind speed, relative humidity and specific humidity. This could have been overlooked easily without considering the d-excess.

Throughout the flight, the aircraft returned four times into this dry air layer above 1700 m a.s.l. (Fig. 10, segments C, D, F, H). Very similar behaviour was observed for the meteorological and SWI parameters at the same altitudes. The profile at segment D is shown in Fig. 9e-h, and underlines the sharpness of the transition at the inversion. From the downward profiles the vertical gradient of the SWI across the inversion is estimated as $-25.4‰\,100\,m^{-1}$ for $\delta D$ and $24.0‰\,100\,m^{-1}$ for the d-excess.

After completion of the vertical profile at Segment A, the aircraft performed a low-level leg at minimum safe altitude (ca. 155 m above the surface) over open water (0712-0725 UTC, segment B). Specific humidity and relative humidity remained narrowly at about $8\,g\,kg^{-1}$ and 57%, respectively during this transect (Table 4a). $RH_{SST}$ was about 57%. The expected non-equilibrium fractionation conditions during evaporation do not fully correspond to the high measured d-excess of ∼37‰ (depending on the non-equilibrium fractionation factor, one would expect here $RH_{SST}$ to be about 18–25% according to the Craig-Gordon model), and point towards evaporation sources of at least part of the sampled water vapour elsewhere. With average wind speeds of $4\,m\,s^{-1}$ from southeasterly directions, only weak evaporation would be expected.

A second low-level transect performed west of Corsica over the open ocean (0805-0825 UTC, segment E) is marked by lower d-excess of 19‰ compared to segment B, which was performed east of Corsica (Table 4b and Figure 10). Specific and relative humidity are slightly higher ($10\,g\,kg^{-1}$ and 65%, respectively) while $RH_{SST}$ was on average 72%, matching to the lower SST (19.5°C) than air temperature (20.6°C). Again, the $RH_{SST}$ does not explain the d-excess, and thus the main sources of the moisture measured by the aircraft may be located elsewhere (in northwesterly direction, as indicated by the mean wind direction of 320°). Throughout the flight, lowest values of the d-excess occur consistently at high RH conditions, at elevations where the BL top is expected, such as during Segment A, C, D and F. This points again to the role of cloud and precipitation evaporation processes in shaping the d-excess minimum observed in the overall profile.

## 7 Airmass origin and transport history

Airmass origin and the transport history for a part of the air sampled during flight 10 is now investigated with a backward trajectory analysis using the LAGRANTO model (Wernli and Davies, 1997) based on ECMWF operational analysis data at





a 0.5°x0.5° horizontal grid spacing. Trajectories were calculated 5 days backward in time every 10 s along the flight track for the time interval 0754–0806 UTC, i.e. during the transition from the free-troposphere into the boundary layer (2nd part of segment D; Fig. 10a). The trajectory analysis confirms a very different origin of the airmasses in the two vertical layers. In the lower troposphere (700–800 hPa), the aircraft encountered airmasses that 4–5 days back in time had been located over

southern Greenland (Fig. 11a, red trajectory segments) in the upper troposphere at pressure altitudes of ∼400 hPa (Fig. 11b, black lines). These airmasses descended gradually by ∼100 hPa day$^{-1}$ before being encountered by the aircraft. Large-scale descent of upper-tropospheric air due to its southward displacement along sloping isentropes with adiabatic warming is thus the most plausible cause for the low relative humidity observed during flight 10. This is also supported by a trajectory analysis for flight segments A, C and F, which show consistent source regions and descent from similar altitudes (not shown).

The trajectories descending from high elevation were very dry (0.3±0.2 g kg$^{-1}$) 4 days before encounter, and hardly moistened according to the ECMWF analysis data at −12 h (0.5±0.4 g kg$^{-1}$). These values agree well with the humidity observed by the aircraft (0.4–0.7 g kg$^{-1}$ during the descend in segment D, Fig. 10c). During descent, potential temperature of the upper tropospheric air masses decreased due to radiative cooling by about 5 K in 5 days and was 303–309 K during encounter (Fig. 11c), in very good agreement with the observations (Fig. 10b). In contrast, the airmass encountered during the lower parts

of segment D was much more humid (6 g kg$^{-1}$, Fig. 11a), lower than the range of the observations (7–10 g kg$^{-1}$). Trajectory analysis shows an origin of these moist airmasses over Italy, France, Corsica, and nearby areas of the Western Mediterranean. The airmasses gradually warmed (Fig. 11c, grey trajectories) and moistened (not shown) while being advected in the lower troposphere from the northwest during the 5 days before encounter (Fig. 11a, blue trajectory segments).

Vertical cross-sections using ECMWF analyses centered at Corsica approximately at the time of flight 10 (0600 UTC 21 Sep

2012) provide further insight into the sharpness of the transition between the free troposphere and the boundary layer in the ECMWF model (Fig. 12). Dry air reached from the upper troposphere down to 750–850 hPa directly over Corsica (upper white areas in Fig. 12a and b), corresponding to the downward sloping isentropic flow from the northwest along the 300 and 305 K isentropes. Further south, a fairly moist (∼5-7 g kg$^{-1}$) airmass with potential temperatures between 305-315 K was located above the dry air, and east of 6°E at higher elevations (above 850 hPa, Fig. 12a). This subtropical airmass associated with

ex-Hurricane Nadine and the cut-off low displaced the dry upper-tropospheric air during the next two days, apparent also as increased cloudiness (Fig. 8c).

Red segments of the flight path during flight 10 mark where the aircraft encountered the very dry upper-tropospheric air, in very good correspondence with the operational analysis. The highest branches of the flight track correspond to the measurements during segments A and D. The less dry conditions, but in particular also the lower depletion of the $\delta$D and $\delta^{18}$O

indicate that the front of the advancing tropical airmass had been encountered during segment D (Fig. 12a), explaining the marked isotopic inversion between 2300 and 3500 m a.s.l. (Figure 9g) as the result of the complex layering of airmasses with very different water vapour transport histories.



## 8 Summary and Conclusions

In summary, we present data from an extensive airborne survey of the SWI composition of water vapour above the western Mediterranean during the HyMeX SOP1. The dataset represents the first airborne characterisation of the SWI composition in the Mediterranean region, and the most extensive measurement of the stable water isotope composition in the lower troposphere since the 1970s. During the 21 successful flights with in total 59 flight hours, several distinct weather situations have been probed, yielding high day-to-day variability and often strong vertical and horizontal gradients in the SWI signals. In general, the variability was much larger than could be anticipated from earlier low-resolution data, a finding confirmed by another recent study (Dyroff et al., 2015). The high variability observed in our data has also implication for remote-sensing observations of stable isotopes and SWI-enabled models. Pronounced horizontal variability is in addition a challenge for model comparison studies that average over larger spatial regions than the aircraft measurements are representative for.

The range of SWI variability is exemplified by a sequence of flights performed in a situation of subsidence of upper and middle troposphere air over the region of operations. Strong vertical gradients in stable water isotopes were found, reaching up to $-25.4‰\,100\,\mathrm{m}^{-1}$ for $\delta$D across an about 700 m thick inversion layer. Mixing between the descending airmass and boundary-layer air influenced by evaporation appears as the dominant process responsible for the observed variability. Immediate, local stable isotope fractionation that could be described by a Rayleigh fractionation model, for example in dense, precipitating clouds, was not observed during the flights, and must thus have taken place elsewhere in regions with saturated conditions beforehand. The capability to reach such conclusions points out the potential of SWI data for evaluating processes in the atmospheric water cycle, delivering information that is not obtainable from water vapour concentration measurements alone.

The airborne d-excess data reported here are the first since Taylor (1972) concluded that the d-excess would not provide additional information. Our unprecedented high-resolution measurements of d-excess clearly provide evidence for the opposite. Careful data processing and calibration is required to obtain interpretable data at a temporal resolution of 15–30 s, and a spatial resolution of 1–2 km horizontally and 75–150 m vertically. Contrary to the conclusions by Taylor (1972), we demonstrate that the d-excess is a useful additional tracer for evaporation processes in the lower troposphere. Elevated d-excess near the sea surface and above the boundary layer inversion are separated from fairly low d-excess near the top of the boundary layer. These characteristics are consistent with varying non-equilibrium evaporation conditions near the surface, evaporation of rain in unsaturated regions and cloud dissolution near the boundary-layer top, and non-linearities in the delta-scale and thus the d-excess, which lead to high values for very depleted conditions. Because of technological challenges, the remaining uncertainties call for further studies to confirm these findings. Nonetheless, the data reported here allow to use the d-excess as a valuable process and source tracer for the considered meteorological situations.

The comprehensive SWI data set acquired together with many other observations during the HyMeX SOP1 (Ducrocq et al., 2014) can be further exploited for process studies using both the primary SWI and the d-excess, as well as for model evaluation. In the future, more extensive data sets for other regions on the atmospheric SWI composition will be highly valuable in constraining the atmospheric water cycle in numerical weather and climate prediction models. Future campaigns would benefit





in particular from complementary sampling of the near-surface water vapour in the evaporation region, either from ship or land-based, providing further insight into mixing between boundary layer and free troposphere.

## Appendix A: Data and measurement details

### A1  Measurement setup

The high-frequency meteorological measurements (temperature, winds, pressure, humidity) onboard the D-IBUF were located inside the nose boom. The inlet for measuring SWIs in ambient air was located at the top of the aircraft at 8.6m distance (Fig. A1). A backward-facing short stainless steel pipe (1/4" O.D.) inside an aerodynamic housing was connected to PTFE tubing (1/4" O.D.) inside the aircraft (Fig. A2). Air was drawn through a PTFE membrane filter and the inlet at $30\,\ell\,\mathrm{min}^{-1}$ at standard conditions using a TF2 pump (Thomas Inc.). At a T-fitting connected to this inlet, the Picarro L2130-i extracted sample air from this air stream at $0.1\,\ell\,\mathrm{min}^{-1}$ using a KNF OEM pump (S2003, Picarro Inc).

### A2  CRDS humidity calibration

The correspondence between the CRDS humidity and the reference humidity allowed to compensate for time shifts between the two measurements due to the inlet, piping, position in the aircraft and computer clock differences. For each flight, the time shift (typically 4–7 s) was determined from the highest correlation between the two humidity measurements, shifted in 1 s intervals within moving 60 s windows. The correlation coefficient between both humidity measurements after time shift correction was $\rho > 0.97$ at a 1 Hz averaging for all 21 flights. Humidity measurements from the CRDS were then calibrated with the HCLY humidity data using a linear fit determined from each individual flight at a 1 Hz time resolution. The slope and offset of this linear fit were quite stable between flights, but changed after the installation of a replacement pump (not shown).

### A3  Humidity dependency correction of the SWI parameters

The humidity-isotope response correction that is required for all commercial water isotope spectrometers is known as a key element for reliable d-excess measurements (Tremoy et al., 2012; Aemisegger et al., 2012). Airborne humidity measurements can have a large range, and instruments need to be calibrated for the humidity dependency across the entire range of measurement data points. Correction functions for the dependency of the SWI parameters were determined at different times during and after the field campaign. These functions are different for each particular analyser. For the instrument used during the HyMeX SOP1, the $\delta^{18}$O and $\delta$D increased strongly for lower mixing ratios, resulting in a strong negative deviation of the d-excess. All $\delta^{18}$O and $\delta$D data were corrected at the highest time resolution before averaging according to the uncalibrated volume mixing ratio reported by the CRDS instrument. The correction functions and their impact on the d-excess are presented in Fig. A3.





## A4 Dependence of data quality on the humidity

As the number of absorbing water molecules decreases in the CRDS cavity with decreasing humidity, the precision of measurements decreases. The uncertainty of the measured sample depends on the ambient water vapour mixing ratio and can be estimated using calibration runs at different water vapour mixing ratios (Figure A5). Depending on the flight and the water vapour mixing ratios the total uncertainty in $\delta$D is $\sim 1.8‰$, for $\delta^{18}$O $\sim 1.0‰$ and for d $\sim 10‰$.

## A5 Pressure effects and hysteresis

Aircraft data reported here were collected up to a pressure altitude of 580 hPa. Cavity pressure and temperatures inside the instrument are continuously stabilised, and records of these parameters indicate that no adjustment problems were encountered during most flight conditions. The instrument did however respond with a time lag to pressure changes inflicted by spiral descent, steep ascent, and turbulent vertical motions. The cavity pressure then deviates up to 0.02 Torr from its set value of 50 Torr (Fig. A5a). This offset is a result of the time delay of the regulating system when stabilising the measurement conditions in the cavity during a pressure change. It could explain a part of the hysteresis seen in Fig. 9 in terms of both water vapour and SWI composition. Interestingly, the d-excess profiles do not show a strong hysteresis effect, which could point towards a compensation of the pressure changes when the d-excess is calculated. After an ascent or descent, the cavity almost immediately returns to the predefined measurement conditions. Herman et al. (2014) also observed a hysteresis of $\delta$D in their measurements and therefore only used downward profiles for satellite validation.

A shift in the linear fit between the water vapour mixing ratio from L2130-i and the D-IBUF fast response measurements (TPLY) can be observed for an upward flight when compared to downward flight (Fig. A5b). Possible causes of this hysteresis are slight shifts in the wavelength monitor or widening of the absorption lines due to observed changes in the cavity pressure (Fig. A5a). This could also affect the isotope measurements. The shifts are small and the effect might be of a 1–2‰ magnitude. Nevertheless this aspect should be kept in mind, when looking at profiles. It is not a priori clear how this affects the d-excess, but the good correspondence of upward/downward profiles of the d-excess indicate a small effect.

To exemplify the CRDS instrument performance during flight, the cavity temperature, cavity pressure and warm box temperature are shown during the same transect of shown in Fig. 10, along with meteorological, SWI and flight track parameters in Fig. A6). After a warmup phase following the power break before start, cavity temperature and warm box temperature stabilize during segment A and remain narrowly within the required specifications throughout the flight. Cavity pressure instead responds immediately to aircraft vertical velocity as expected from Fig. A6b. During a period of increased turbulence in segment G vertical winds increase, but do not show a visible impact on the instrument parameters. The SWI parameters $\delta$D and d-excess do not show clear indications of the cavity pressure changes, as the environmental signals are exceeding the effect of pressure changes substantially.

SWI calibration was routinely carried out before and after flights to test the stability of the instrument and to correct for a continuous drift of the wavelength monitor during the campaign. While calibration during flight may be desirable to check that the instrument does not perform random jumps during the 3–4 h of flight time, the parallel measurement of water vapour





($H_2^{16}O$) with other on-board instrumentation provides a valid control that the instrument did in fact remain stable during normal flight situations, while the calibrations bracketing each flight ensured the same for the SWI measurements. Thus, calibration during flight would not improve the data quality substantially in our case, but rather have substantially reduced the available measurement time.

5 *Acknowledgements.* We thank Kate Dennis (Picarro Inc.) for helpful support in operating the custom-modified stable water isotope analyser. B. Adler (IMG-TRO) kindly helped extracting the data from the energy balance station at San Giuliano, Corsica. F.A. was partly supported by a Swiss National Science Foundation (SNSF) grant (P2EZP2_155603).





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



**Table 1.** Instrumentation onboard the Do 128-6 during HyMeX SOP1

| Parameter | Measurement principle | Type and manufacturer | Measurement frequency |
|---|---|---|---|
| Specific humidity | Dew point mirror | TP3, Meteolabor | variable |
| | Lyman-$\alpha$ | Buck Research | 100 Hz |
| | Humicap hygrometer | HMP 233, HumiCap | 100 Hz |
| | CRDS Laser spectrometer | L2130-i, Picarro Inc. | 1 Hz |
| $\delta$D, $\delta^{18}$O | CRDS Laser spectrometer | L2130-i, Picarro Inc. | 1 Hz |
| Temperature | PT100 open-wire | Rosemount | 100 Hz |
| Pressure | 5-hole probe and pressure transducers | Rosemount | 100 Hz |
| Surface temperature | Infrared radiometer | Heiman KT15 | 1 Hz |
| Wind speed | calculated from flight parameters | | 100 Hz |





**Table 2.** Flight dates and corresponding pattern during the HyMeX D-IBUF deployment. During the flights printed in italic the inlet pump of the stable isotope measurements was not working correctly.

| Flight number | Date | Takeoff (UTC) | Landing (UTC) | Duration (h) | Pattern |
|---|---|---|---|---|---|
| 01 | 11 Sep 2012 | 11:01 | 14:00 | 2:59 | 1 |
| 02 | 12 Sep 2012 | 07:59 | 11:05 | 3:06 | 5 |
| 03 | 12 Sep 2012 | 14:07 | 17:19 | 3:12 | 5 |
| 04 | 13 Sep 2012 | 07:56 | 10:45 | 2:49 | 5 |
| 05 | 13 Sep 2012 | 13:07 | 13:47 | 0:40 | 2 |
| 06 | 14 Sep 2012 | 08:00 | 11:06 | 3:06 | 5 |
| 07 | 17 Sep 2012 | 08:59 | 12:01 | 3:02 | 1 |
| 08 | 18 Sep 2012 | 08:59 | 12:01 | 3:02 | 1 |
| 09 | 20 Sep 2012 | 07:53 | 10:53 | 3:00 | 3 |
| 10 | 21 Sep 2012 | 06:56 | 09:58 | 3:02 | 3 |
| 11 | 21 Sep 2012 | 11:23 | 14:26 | 3:03 | 3 |
| 12 | 23 Sep 2012 | 06:55 | 10:03 | 3:08 | 1 |
| 13 | 23 Sep 2012 | 11:00 | 14:07 | 3:07 | 1 |
| 14 | 24 Sep 2012 | 11:56 | 15:14 | 3:18 | 9 |
| 15 | 25 Sep 2012 | 07:01 | 10:09 | 3:08 | 1 |
| 16 | 25 Sep 2012 | 11:09 | 14:11 | 3:02 | 1 |
| *17* | *27 Sep 2012* | *12:54* | *16:20* | *3:26* | *9* |
| *18* | *28 Sep 2012* | *10:00* | *13:18* | *3:18* | *TES* |
| *19* | *28 Sep 2012* | *14:05* | *15:35* | *1:30* | *1* |
| *20* | *1 Oct 2012* | *08:55* | *11:30* | *2:35* | *2* |
| *21* | *1 Oct 2012* | *13:04* | *16:07* | *3:03* | *2* |
| *22* | *2 Oct 2012* | *08:58* | *12:09* | *3:11* | *1* |
| *23* | *2 Oct 2012* | *13:39* | *16:32* | *2:53* | *1* |
| *24* | *3 Oct 2012* | *08:50* | *12:01* | *3:11* | *3* |
| *25* | *3 Oct 2012* | *13:09* | *16:33* | *3:24* | *3* |
| *26* | *4 Oct 2012* | *08:42* | *12:10* | *3:28* | *5* |
| 27 | 4 Oct 2012 | 12:59 | 15:53 | 2:54 | 7 |
| 28 | 5 Oct 2012 | 07:31 | 10:36 | 3:05 | IASI |
| 29 | 5 Oct 2012 | 13:09 | 14:38 | 1:29 | 3 |
| 30 | 9 Oct 2012 | 08:57 | 11:37 | 2:40 | 1 |
| 31 | 10 Oct 2012 | 11:35 | 13:55 | 2:20 | 3 |
| 32 | 11 Oct 2012 | 06:36 | 10:03 | 3:27 | Falcon |



**Table 3.** Relation between averaging times and standard deviations, and the resulting vertical and horizontal resolution at typical true air speed of $65\,\mathrm{m\,s^{-1}}$ horizontally and $5\,\mathrm{m\,s^{-1}}$ vertically at a volume mixing ratio of 9000 ppmv.

| Averaging time (s) | 5 | 10 | 15 | 30 | 60 | 180 |
|---|---|---|---|---|---|---|
| $\delta D$ (‰) | 2.47 | 2.39 | 2.36 | 2.33 | 2.32 | 2.33 |
| $\delta^{18}O$ (‰) | 0.50 | 0.43 | 0.40 | 0.38 | 0.36 | 0.35 |
| d-excess (‰) | 3.17 | 2.29 | 1.91 | 1.47 | 1.18 | 0.93 |
| Horizontal resolution (m) | 325 | 650 | 975 | 1950 | 3900 | 11700 |
| Vertical resolution (m) | 25 | 50 | 75 | 150 | 300 | 900 |

**Table 4a.** Mean and standard deviation of meteorological and SWI parameters measured during the low-level transects over open water during flights 09–11 as part of flight pattern #3. Values are mean and standard deviation of 15 s mean data of the parameters WD (wind direction), WS (wind speed), and RH (relative humidity).

| Flight | Time (UTC) | Segment | GPS Altitude (m a.s.l.) | WD (°) | WS ($\mathrm{m\,s^{-1}}$) | $T_{air}$ (°C) | q | RH (%) |
|---|---|---|---|---|---|---|---|---|
| 9 | 08:08:50-08:19:20 | B | 154.3±4.1 | 39.1±48.0 | 6.4± 0.5 | 20.7± 0.1 | 12.0± 0.2 | 77.1±1.5 |
| 9 | 09:00:40-09:17:35 | E | 156.3± 3.6 | 225.9±14.3 | 4.0± 1.0 | 19.3± 0.3 | 9.6± 0.3 | 67.0±2.8 |
| 10 | 07:15:35-07:24:35 | B | 155.0± 2.6 | 103.5±21.5 | 4.1± 1.0 | 19.2± 0.1 | 8.0± 0.4 | 56.7±3.2 |
| 10 | 08:06:10-08:23:25 | E | 155.0± 3.2 | 320.0± 8.7 | 5.2± 1.2 | 20.6± 0.3 | 10.0± 0.6 | 65.4±4.2 |
| 11 | 11:42:50-11:51:35 | B | 156.5± 1.9 | 121.1± 7.9 | 3.9± 0.3 | 19.5± 0.1 | 9.6± 0.4 | 66.7±3.0 |
| 11 | 12:34:10-12:51:25 | E | 156.9± 3.5 | 269.5±41.2 | 4.0± 1.4 | 21.3± 0.4 | 9.4± 0.7 | 58.3±5.5 |

**Table 4b.** As Table 4a, but for the parameters $RH_{SST}$ (relative humidity with respect to SST), SST, and the SWI parameters $\delta^{18}O$, $\delta D$, and d-excess.

| Flight | Time (UTC) | Segment | GPS Altitude (m a.s.l.) | $RH_{SST}$ (%) | SST (°C) | $\delta^{18}O$ (‰) | $\delta D$ (‰) | d-excess (‰) |
|---|---|---|---|---|---|---|---|---|
| 9 | 08:08:50-08:19:20 | B | 154.3±4.1 | 56.8±2.2 | 26.4± 0.6 | -12.2±0.5 | -82.4± 1.8 | 15.0± 4.5 |
| 9 | 09:00:40-09:17:35 | E | 156.3± 3.6 | 48.4±2.7 | 25.2± 0.9 | -14.7± 0.9 | -90.5± 4.1 | 27.3± 6.1 |
| 10 | 07:15:35-07:24:35 | B | 155.0± 2.6 | 56.6±3.7 | 19.7± 0.4 | -16.5± 1.3 | -95.1± 8.1 | 36.5± 7.7 |
| 10 | 08:06:10-08:23:25 | E | 155.0± 3.2 | 72.2±5.6 | 19.5± 1.0 | -13.6± 0.5 | -90.1± 2.6 | 19.0± 3.6 |
| 11 | 11:42:50-11:51:35 | B | 156.5± 1.9 | 65.6±3.3 | 20.3± 0.4 | -13.1± 0.5 | -84.8± 1.5 | 19.7± 3.6 |
| 11 | 12:34:10-12:51:25 | E | 156.9± 3.5 | 63.6±7.6 | 20.5± 1.7 | -13.0± 0.5 | -89.1± 1.6 | 14.6± 3.8 |





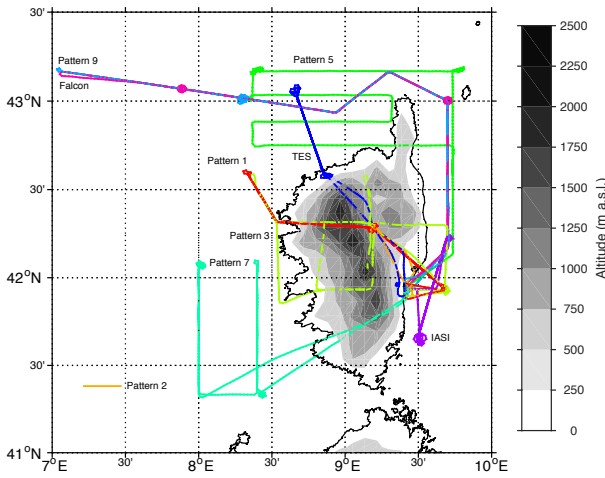

**Figure 1.** Flight patterns during the HyMeX IBUF campaign. Grey shading shows elevation in m above sea level (m a.s.l.). Table 1 lists the dates when the different patterns were flown.

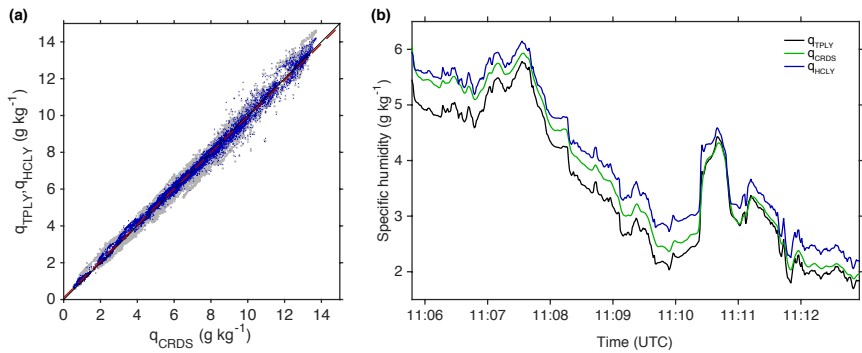

**Figure 2.** (a) Correlation between calibrated specific humidity from the CRDS ($q_{CRDS}$), the humicap/Lyman-$\alpha$ ($q_{HCLY}$, blue dots, $\rho = 0.994$), and the dewpoint hygrometer/Lyman-$\alpha$ product ($q_{TPLY}$, grey dots, $\rho = 0.989$) at 1 s time interval for flight 1 on 11 Sep 2012. (b) Comparison of calibrated humidity measurements from the three products for a representative vertical profile during flight 1 ascending from 1500 m to 3700 m a.s.l.

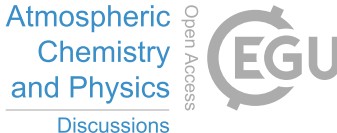

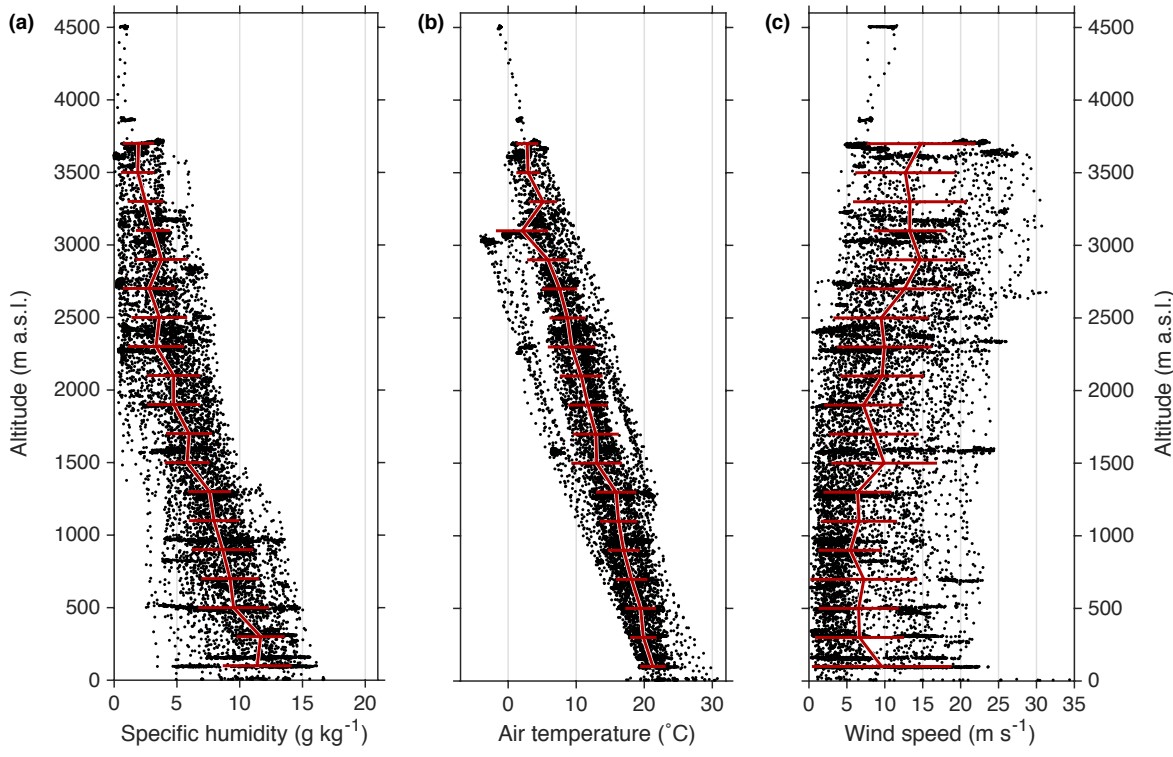

**Figure 3.** Vertical profiles of the mean thermodynamic and kinematic state of the atmosphere observed during the campaign. (a) Specific humidity from HCLY in $g\,kg^{-1}$, (b) air temperature in K, (c) horizontal wind speed in $m\,s^{-1}$. Overlaid are the 200 m-binned averages as red lines. Horizontal red bars denote the bin 1-$\sigma$ standard deviation.





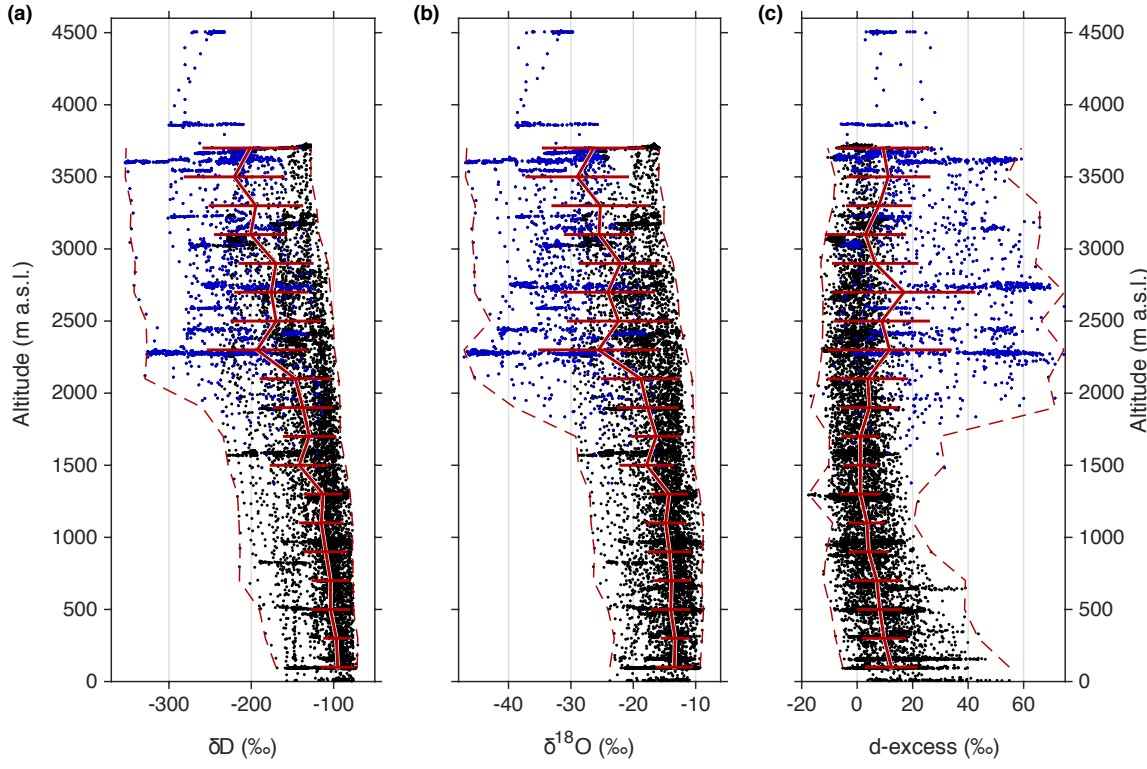

**Figure 4.** Vertical profiles of the SWI composition of ambient water vapour for all valid flight data acquired during the campaign. Overlaid are 200 m-binned mean (red solid line), and 200 m-binned minimum and maximum values (dashed red lines) of (a) $\delta$D, (b) $\delta^{18}$O, and (c) d-excess. Data points with a specific humidity below $2\,\mathrm{g\,kg^{-1}}$ are shown in blue. Bin averages do not extend above 3700 m because of low data coverage.



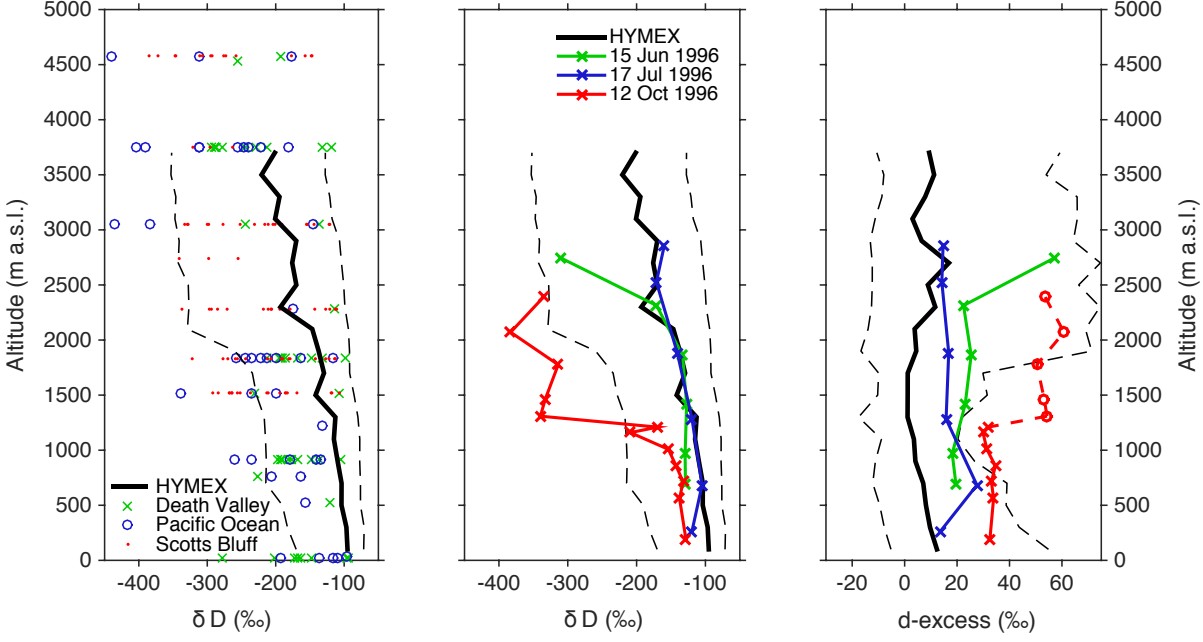

**Figure 5.** (a) Comparison of HyMeX measurements of $\delta$D (mean and range, black lines) to airborne $\delta$D data by Ehhalt (1974); Ehhalt et al. (2005) from repeated flights over Death Valley, California (green crosses), the Pacific Ocean off the coast of California (blue circles), and Scottsbluff, Nebraska (red dots). (b) Comparison of HyMeX measurements of $\delta$D and (c) d-excess (mean and range, black lines) to flight measurements by He and Smith(1999) over New England forests on 15 June 1996 (green), 17 July 1996 (blue), and 12 Oct 1996 (red). Red circles and dashed lines denote data points for which He and Smith (1999) estimated the d-excess from all other measurements due to low humidity conditions.



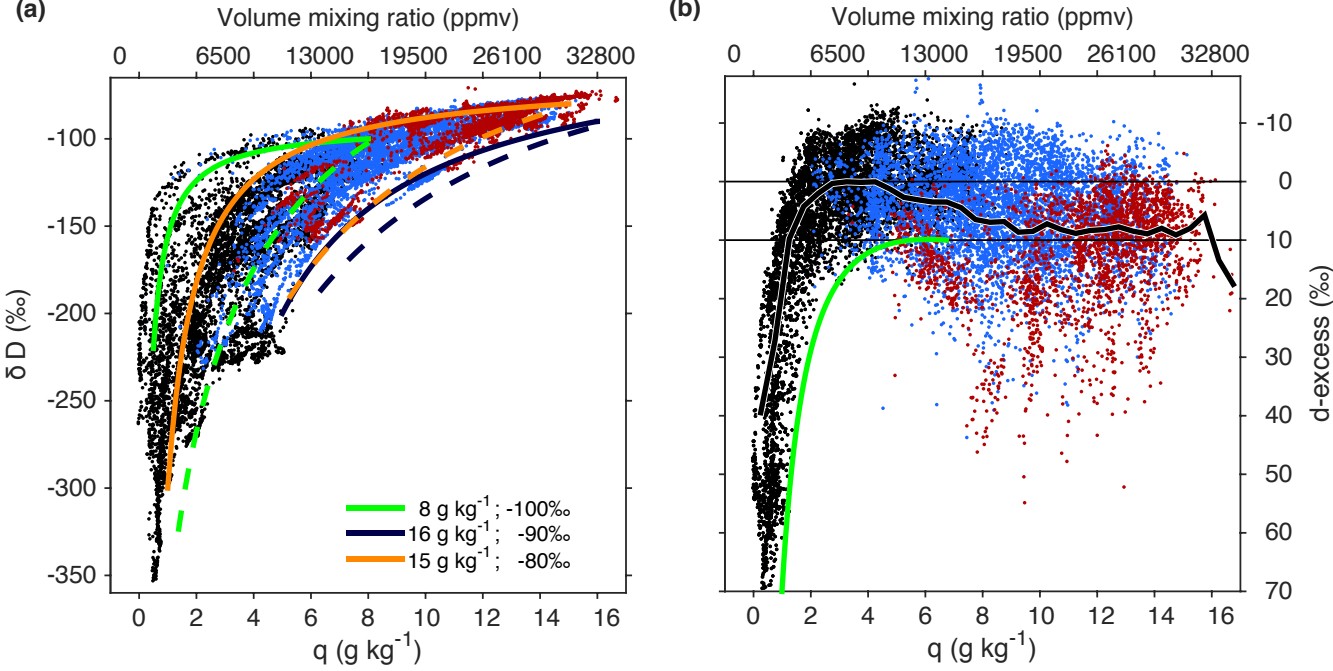

**Figure 6.** Scatter diagrams of (a) δD (‰) and (b) d-excess (‰) versus specific humidity q (g kg$^{-1}$). Shown are 30 s averaged data from all valid flights. Data points acquired below 500 (1700) m a.s.l. are marked in red (blue). Colored curves in (a) represent idealised mixing (solid lines) and Rayleigh fractionation processes (dashed lines) as described in the main text. Upper x-axis gives water concentration in units of ppmv. Green line in (b) represents d-excess from a Rayleigh fractionation process. Black line in (b) is a bin average. Note that the d-excess axis in (b) is inverted for display purposes.

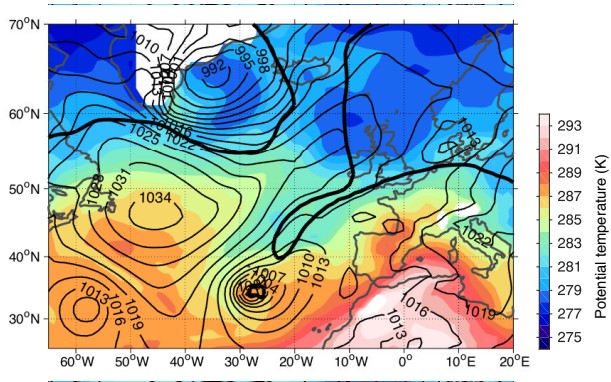

**Figure 7.** Weather situation on 12 UTC 21 Sep 2012 (flight 09 to 11). Shown are sea level pressure (black lines, 3 hPa contour interval), equivalent potential temperature at 850 hPa (color shading, 1 K interval) and the tropopause structure depicted by a contour of 2 potential vorticity units on the 315 K isentrope (thick black line) from ECMWF analysis data.

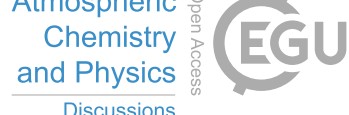



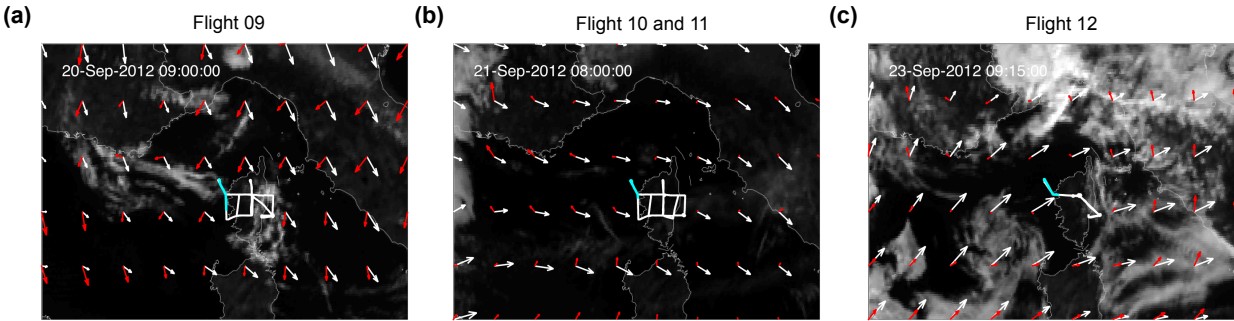

**Figure 8.** Cloudiness and winds at (a) 09 UTC 20 Sep 2012 during flight 09, (b) 09 UTC 21 Sep 2012 during flight 10 and 11, and (c) 09 UTC 23 Sep 2012 during flight 12. Shown are Meteosat 9 infrared imagery with flight track (white line) and aircraft position during the profiles (cyan line), and ECMWF analysis winds at 500 hPa (white arrows) and 925 hPa (red arrows) overlayed.





**Figure 9.** Vertical profiles of air temperature (T, °C, first column), specific humidity (q, g kg$^{-1}$, second column), $\delta$D (‰, third column) and the d-excess (‰, fourth column), for flights 09 to 12 conducted during 21 to 23 Sep 2012. Downward profiles are less affected by memory (black or solid lines) than upward profiles (grey or dashed lines). Red lines in panels (i) to (l) show the downward profile acquired during flight 12.



**Figure 10.** Segments A to H of flight 10 on 21 Sep 2012, 0700–0925 UTC. (a) flight altitude (black line) and topography (grey shading), (b) potential temperature (K, black) and air temperature (°C, red), (c) specific humidity from CRDS (g kg$^{-1}$, black) and the HCLY product (g kg$^{-1}$, red stippled), (d) relative humidity from HCLY temperature and humidity (%, black), (e) wind speed (m s$^{-1}$, black) and wind direction (°, red), (f) $\delta$D (‰, black) at 15 s averaging time, (g) $\delta^{18}$O (‰, black) at 15 s averaging time, (h) d-excess (‰) at 15 s averaging time (grey) and 30 s averaging time (black). Sections highlighted in red are for specific humidity below 2 g kg$^{-1}$. Labels A–H denote different sections of the flight pattern.





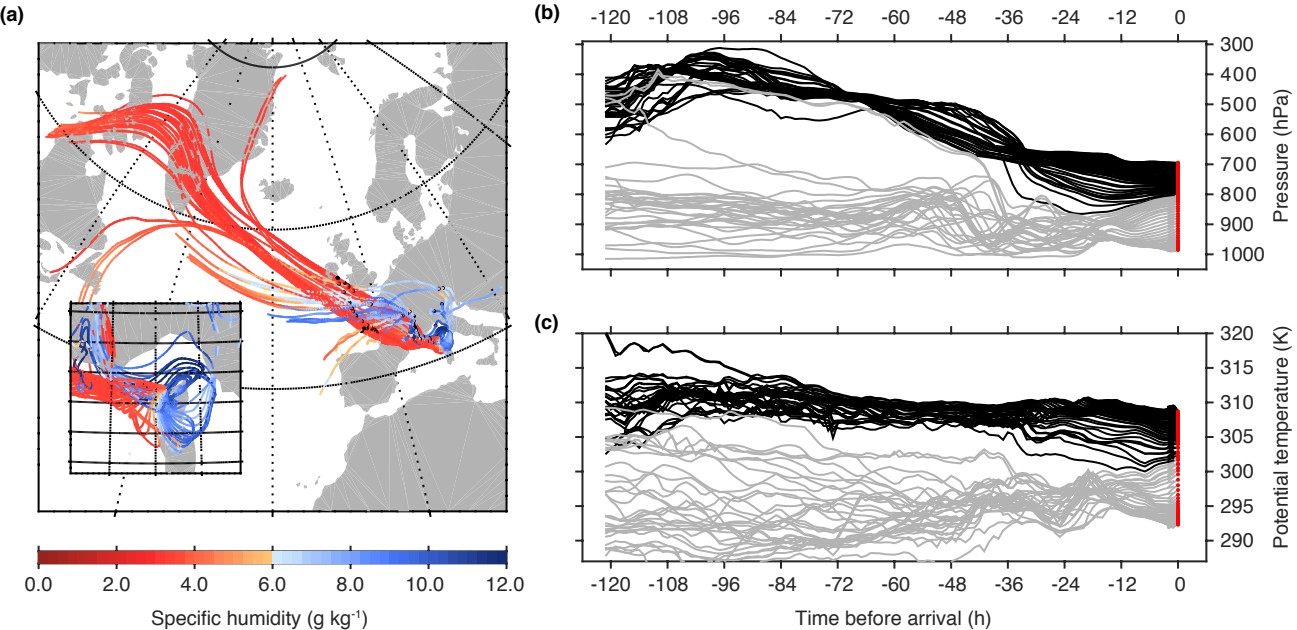

**Figure 11.** Backward trajectories for the downward leg of the second profile in segment D during flight 10 (21 Sep 2012, 0754–0806 UTC). (a) 5-day trajectories colored with specific humidity ($g\,kg^{-1}$), with small inset showing a zoom around Corsica. Black dots indicate trajectory locations 48 h before measurement time. (b) Pressure in hPa and (c) potential temperature in K along 5-day trajectories vs. time before measurement. Trajectories arriving with a humidity of less than $2\,g\,kg^{-1}$ are shown in black, all others in grey. Time of aircraft encounter in panel (b) and (c) are marked by red dots.





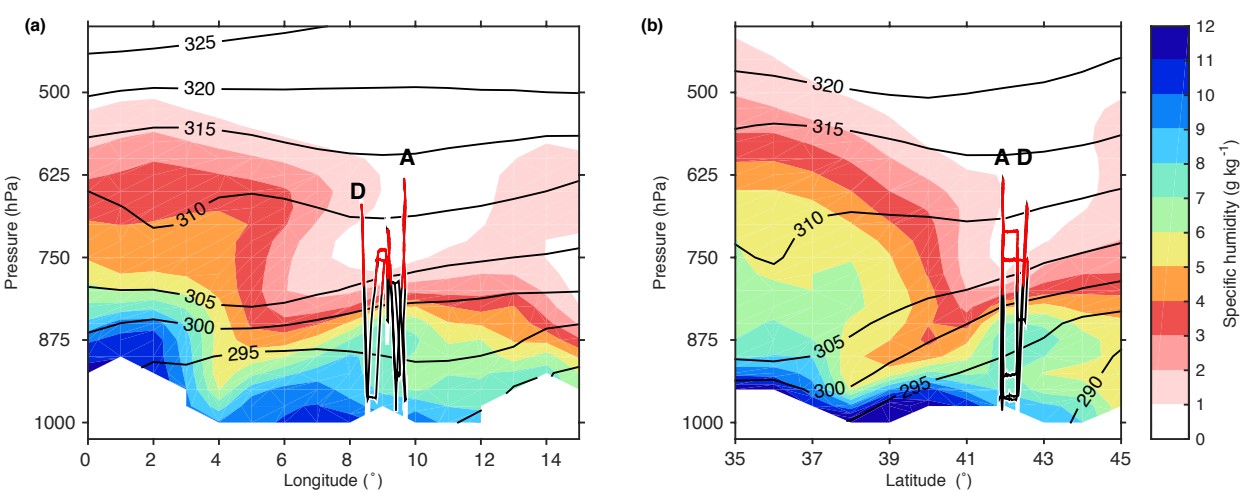

**Figure 12.** Vertical sections across Corsica during flight 10 (0600 UTC 21 Sep 2012), (a) from 0–15°E at 42°N, and (b) from 35–45°N at 9°E. Panels show specific humidity (g kg$^{-1}$, shading), and potential temperature (K, black contours) from ECMWF operational analyses. The flight track for flight 10 is overlaid as black line, with segments of specific humidity below 2 g kg$^{-1}$ plotted in red. A and D correspond to the segments in Fig. 10.





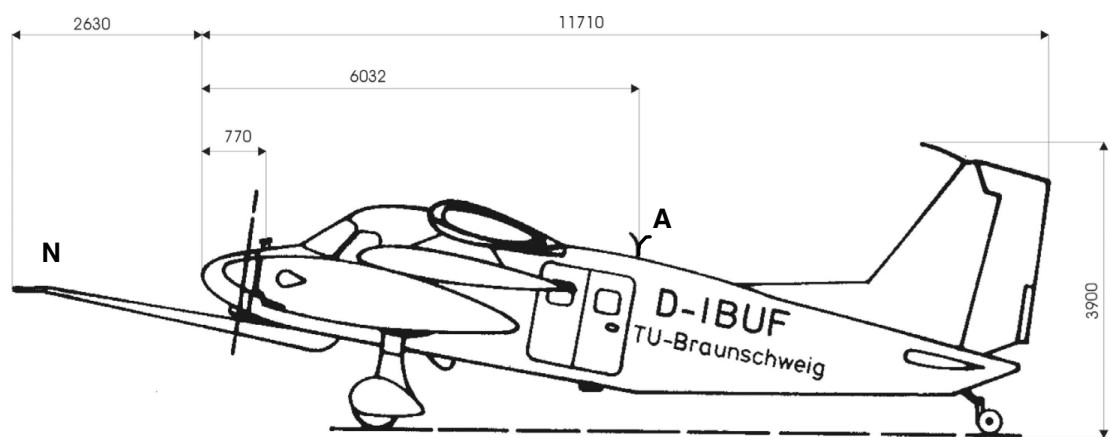

**Figure A1.** Location of the nose boom instrumentation (N) and the inlet for the ambient air measurements (A) onboard the Dornier-128 D-IBUF aircraft aircraft for the HyMeX measurement campaign. All dimensions in mm. Modified from Wieser (2005).





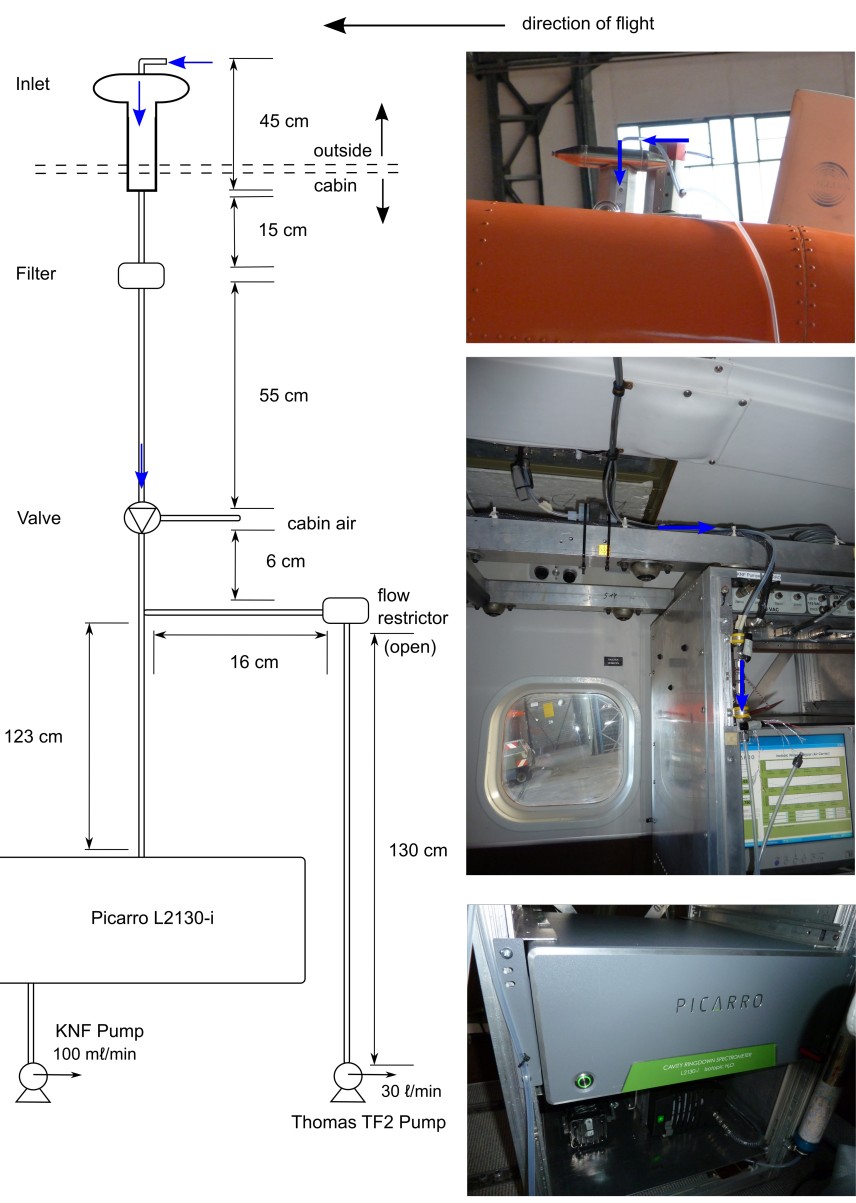

**Figure A2.** Setup of the Picarro L2130-i instrument on-board the Dornier-128 D-IBUF. The pipe connected to the inlet in the top right is not part of the inlet system. From Aemisegger (2013).





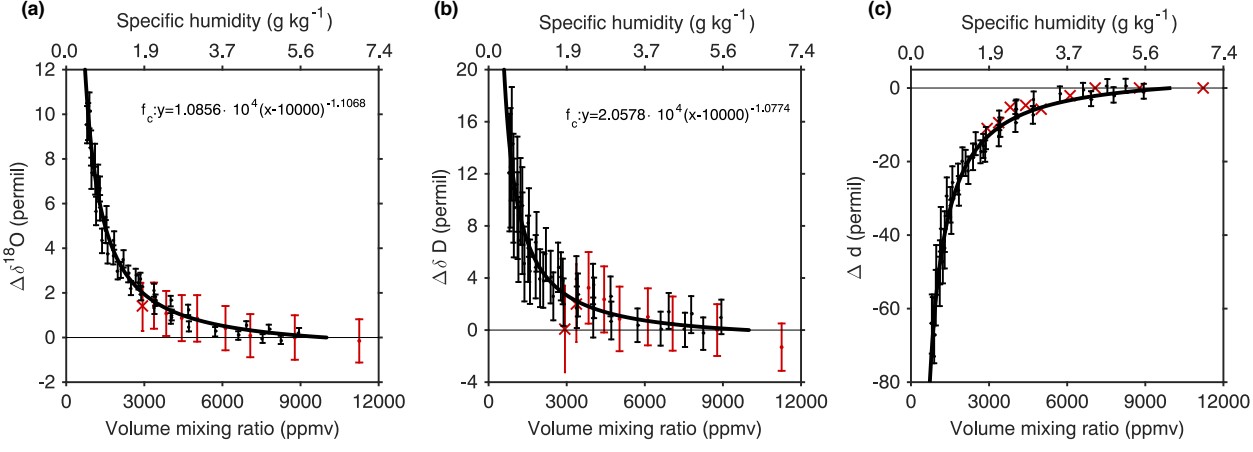

**Figure A3.** Correction functions of the SWI composition as a function of the water concentration (volume mixing ratio in ppmv) determined during laboratory experiments after the field campaign for (a) $\delta^{18}O$, (b) $\delta D$. (c) Effect on the d-excess when applying the correction to the primary stable isotope parameters. Red data points were obtained in 2012, black data points in 2015.

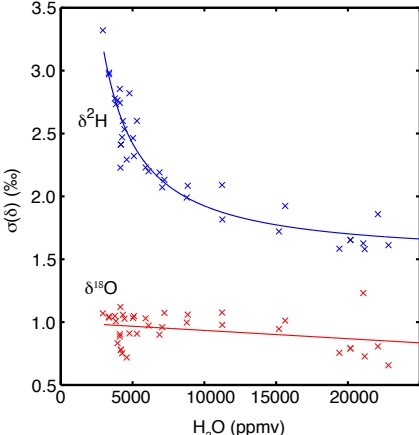

**Figure A4.** Standard deviation of calibration standard measurements in the lab at different water vapour mixing ratios (experiments from 26 Sep 2012 and 25 Oct 2012).



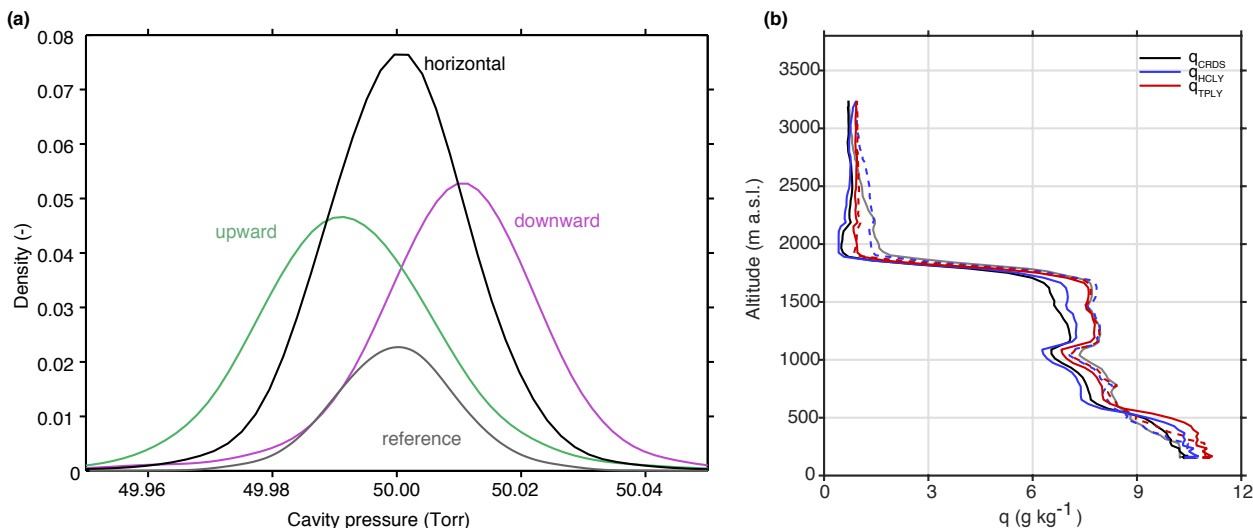

**Figure A5.** (a) Distributions of cavity pressure for flights 1–14 and 28–32 in red for vertical velocities $< 2\,\mathrm{m\,s^{-1}}$ (downward), in blue for vertical velocities $> 2\,\mathrm{m\,s^{-1}}$ (upward), in black for horizontal legs with vertical velocites $2 \leq v \leq 2\,\mathrm{m\,s^{-1}}$. The distribution of the cavity pressure during a stability experiment (Allan test) in the hangar during the campaign (06 Oct 2012 10–14 UTC) is shown in grey as a reference for stable measurement conditions. (b) Vertical profile of specific humidity during flight 10. Upward (dashed lines) and downward (solid lines) from three humidity products CRDS (black/grey), HCLY (blue) and TPLY (red) are shown.





**Figure A6.** Instrument control parameters during segments A to H of flight 10 on 21 Sep 2012, 0700–0925 UTC. (a) Flight altitude (m a.s.l., black), cabin pressure (hPa, red) and topography (m, grey shading), (b) vertical wind (m s$^{-1}$), (c) CRDS cavity pressure (Torr, black) and vertical aircraft velocity (m s$^{-1}$, red), (d) CRDS cavity temperature ($^{\circ}$C, black) and CRDS warm box temperature ($^{\circ}$C, red), (e) $\delta$D (black line, ‰) at 15 s averaging time, (f) d-excess (‰) at 15 s averaging time (grey line) and 30 s averaging time (black line). Sections highlighted in red are for specific humidity below 2 g kg$^{-1}$.