# Peer review of "The stable isotopic composition of water vapour above Corsica during the HyMeX SOP1 campaign: insight into vertical mixing processes from lower-tropospheric survey flights"

_Atmospheric Chemistry and Physics, 2016_

## Referee Comment (RC1) · Anonymous Referee #1 · 18 Oct 2016

Review of

*"The stable isotope composition of water vapour above Corsica during the HyMeX SOP1: insight into vertical mixing processes from lower-tropospheric survey flights"*

by H. Sodemann et al.

Paper published in ACPD on 4 October 2016

**General comments** This paper presents airborne in situ measurements of the stable water isotope (SWI) ratios $\delta D$, $\delta^{18}O$, and the second order isotope parameter deuterium excess from 21 flights over the Mediterranean. Thanks to the relatively high temporal resolution of the measurements, the study is able to reveal highly variable and well-structured SWI profiles as well as a very variable relation between specific humidity and SWI ratios. The mean values and the variability of the vertical SWI profiles are in good agreement with sparse existing airborne SWI measurements and are discussed with respect to condensation, air mass mixing, and processes related to evaporation. Based on the evolution of the SWI ratios during four days the authors find that both large-scale transport and vertical mixing within separated layers are responsible for the observed variable and well-structured vertical SWI profiles. This is supported by a backward trajectory analysis, which confirms the relation between the observed SWI ratios and air mass origin and transport.

Especially the observations of deuterium excess are unique with respect to the amount of data and temporal resolution and of high interest for atmospheric studies on transport and isotope microphysics and the validation of isotope-enabled models. The data is complemented by a comprehensive and convincing interpretation. I recommend publication of this innovative, very well-written paper after my following major two concerns regarding measurement uncertainty have been addressed:

1. In my opinion, a central assumption of this publication regarding the measurements of deuterium excess is stability of the deuterium excess-humidity response of the instrument. The applied deuterium excess-humidity response correction was determined after the campaign. The fact that this response is individual for each Picarro Isotope Analyzer may indicate that the response is related to the alignment of optical components or other parameters that could change with time or in consequence of mechanical stress. Because the magnitude of the applied humidity correction for deuterium excess (Fig.A3c) is of the same order of magnitude as the observed range of observations of deuterium excess (Fig.6b) this may introduce significant systematic uncertainty to the observations. The authors mention additional calibrations of the SWI-humidity response during the campaign. Could you add them to Fig.A3 to demonstrate stability of the SWI-humidity response correction?

2. Uncertainty estimates for the SWI measurements are based on the standard deviation of calibration standard measurements in the lab (Fig.A4). However, this estimate does not account for (a) memory effects of the inlet tubing and other effects responsible for the observed hysteresis of SWI measurements, (b) potentially increased instrumental drift in consequence of mechanical stress during the flights, and (c) systematic uncertainty of the applied SWI-humidity response correction. Could you quantify these three uncertainty terms e.g. for humidity levels of 1000, 5000, and 10000 ppmv:

- (a) The observed hysteresis of SWI measurements should be consistent with the stated measurement uncertainty. Could you quantify the observed mean hysteresis of the SWI measurements for different humidity levels?

  In my opinion, not only changes in cavity pressure but also memory effects of inlet tubing and analyzer are reasonable explanations for the observed strong hysteresis of the SWI measurements. The 123 cm of inlet tubing closest to the isotope analyzer were not flushed by the TF2 flushing pump. Considering the small air stream of 0.1 l min$^{-1}$ through this part of the tubing, the relatively strong adsorption of water on PTFE surfaces, and encountered strong gradients of specific humidity memory effects may play an important role. This seems to be supported by the differences between SWI measurements during ascent and descent, which appear to be especially pronounced during dry conditions and smaller during moist conditions. Compare e.g. Fig9g: dry conditions above 1700 m a.s.l. are related to a strong hysteresis of $\delta D$, wet conditions below 1700 m a.s.l. are related to a small hysteresis of $\delta D$.

  With respect to memory effects of the measurement system the authors state time constants of 2.4 s, 1.7 s, and 1.3 s for $\delta D$, $\delta^{18}O$, and $H_2^{16}O$, which are within the applied averaging time of 15 s. These time constants were determined in Aemisegger [2013]. As I understand it, Aemisegger [2013] derived the time constants by measuring the instrument response to changes of specific humidity of 1000 ppmv between 12000 and 22000 ppmv. I assume that these time constants may be substantially larger for drier conditions and larger humidity steps e.g. from 5000 to 500 ppmv.

- (b) During the campaign SWI calibrations were performed before and after the flights. However, most of the calibrations before the flights had to be discarded because stabilization of the instrument was not completed. Could you estimate typical drift of the isotope analyzer during the flights based on the few days with successful calibrations before and after a flight?

- (c) Do the error bars in Fig.A3 state the total uncertainty of the SWI-humidity response calibration? Could you underpin stability of the SWI-humidity response calibration by measurements during the campaign?

**Specific comments**

- p.2,L.8: In my opinion, stating the definition of the $\delta$-notation or citing e.g. [Coplen, 2011] would be helpful.

- p.6,L.1: "...changed after the installation of a replacement pump..." Slope and offset between CRDS humidity measurements and HCLY measurements were stable between flights, but changed after the installation of a replacement pump. Does this change indicate memory effects of the inlet tubing, whereby the impact of these effects changed in magnitude in consequence of an adjusted air flow?

- p.6,L.10-28: See my first point in the general comments.

- p.6,L.30-p.7,L.10.: SWI measurements with the Picarro isotope analyzer are based on linear two-point calibrations at $\delta D = -78.68‰$ and $-166.74‰$. Aemisegger et al. [2012] show significant non-linearity of this correction for an older version (L1115-i) of the Picarro isotope analyzer. As many of the observations with high d-excess are related to $\delta D$ ratios smaller than $-166.74‰$ (Fig.4) it might be of interest to quantify respective additional measurement uncertainty at small isotope ratios.

- p.7.,L.15-L.18: The required measurement accuracies of $H_2^{16}O$, $\delta D$, and especially of deuterium excess are of different order of magnitude. A drift of the CRDS $H_2^{16}O$ calibration by 10‰ would be almost undetectable if comparing the CRDS and HCLY humidity measurements. However, changing $\delta D$ and $\delta^{18}O$ by 10‰ would shift the deuterium excess even for 70‰. I therefore think that a small observed drift of the CRDS $H_2^{16}O$ calibration does not necessarily confirm a small drift of the SWI calibration.

- p.7,L.22-24: Dyroff et al. [2015] used a different measurement principle. For this reason please skip this sentence.

- p.7,L.26-28: "From laboratory experiments with the inlet system..." Please add a description of this experiment to the Appendix. See my point 2a in the general remarks.

- p.9,L.33: Could you explain in more detail what you mean by non-linearity of the delta scale.

- p.9,L.31-p.10,L.1: "The high d-excess ... is therefore not an indication of insufficient data quality, but a real feature ... in the atmosphere": I agree that the high d-excess encountered is not an indication of insufficient data quality, but I don't think that agreement with the observations of Galewsky et al. [2011] and Samuels-Crow et al. [2014] alone justifies the conclusion that the SWI composition presented here is a real feature. I would rather treat this agreement as further evidence for reliable observations. Please rephrase this sentence.

- p.12,L.6: "a very dry and depleted free tropospheric air mass": The free tropospheric end member you assume for the green mixing curve is drier but less depleted than the free tropospheric end member corresponding to the orange curve. Can the free tropospheric end member corresponding to the green mixing curve be explained by condensation alone or do the specific humidity of $0.5\,g\,kg^{-1}$ and the $\delta D$ of $-220‰$ imply earlier mixing?

- p.14,L.34: As mentioned above I would appreciate a quantification of the memory effects regarding uncertainty of the SWI measurements.

- p.15,L.33-34: "Reproducibility was good ... except for the dry intermediate layer". To me this seems like further evidence for a humidity-dependent and not a cavity pressure-dependent hysteresis of the SWI measurements.

- p.20,A1: Was temperature in the non-pressurized cabin stabilized? Did you heat the tubing to avoid condensation?

- p.21,L.3-5: I wouldn't use the standard deviation of calibration standard measurements as measure for the total uncertainty of the observations during the campaign as this value doesn't consider the effects causing the observed hysteresis of SWI measurements, potentially increased instrumental drift during the flights in consequence of mechanical stress, and systematic uncertainty of the humidity response correction. See my second point in the general remarks.

- p.21,L.18-20: "Possible causes of this hysteresis... due to observed changes in the cavity pressure": What about memory effects causing the hysteresis?

- p.21,L.32-p.22,L.4: The authors refer to SWI calibrations bracketing each flight ensuring stability of the SWI measurements. This is in contrast to p.7,L.6-7: "...with very few exceptions only the calibration runs after the daily flight operations were used...". Maybe you could add a figure demonstrating the stability of SWI measurements based on the few days with successful calibrations before and after the flights. Otherwise I would skip the sentence p.22,L.2-4 "...calibration during flight would not improve data quality substantially in our case..."

- Fig2a: What is the meaning of the red and black lines.

- Fig2,Fig3,Fig6,Fig9,Fig10,Fig11,Fig12,FigA3,FigA4,FigA5: I would appreciate consistent use either only of ppmv or only of g kg$^{-1}$ for stating humidity in the different figures and in the text.

- Fig6b: Could you add systematic uncertainty of the deuterium excess measurements (resulting e.g. from uncertainty of the SWI-humidity response correction) to this figure?

- Fig.A3: Do the error bars show statistical uncertainty from measurement noise? What is the meaning of the red crosses?

**Technical corrections**

- p.7.,L.30: CRDS

- p.8,L.4: "1.17": Table 3 states an uncertainty of 1.18‰ for d-excess.

- p.11,L.31: "relatively moist": Do you mean relatively dry?

- p.11,L.35: "magenta line": The respective line is orange.

- p.21,L.4: "Figure A5": Figure A4?

- Fig9: "black or solid ... grey or dashed lines": There are no dashed lines visible in this figure.

- Fig10: "red stippled": not shown

- Fig.A5: "red": magenta; "blue": green

F. Aemisegger. *Atmospheric stable water isotope measurements at the timescale of extratropical weather systems.* PhD thesis, ETH Zuerich, 2013.

F. Aemisegger, P. Sturm, P. Graf, H. Sodemann, S. Pfahl, A. Knohl, and H. Wernli. Measuring variations of $\delta 18O$ and $\delta 2H$ in atmospheric water vapour using two commercial laser-based spectrometers: an instrument characterisation study. *Atmospheric Measurement Techniques*, 5(7):1491–1511, jul 2012. ISSN 1867-8548. doi: 10.5194/amt-5-1491-2012. URL http://www.atmos-meas-tech.net/5/1491/2012/.

T. B. Coplen. Guidelines and recommended terms for expression of stable-isotope-ratio and gas-ratio measurement results. *Rapid Communications in Mass Spectrometry*, 25(17):2538–2560, sep 2011. ISSN 09514198. doi: 10.1002/rcm.5129. URL http://doi.wiley.com/10.1002/rcm.5129.

C. Dyroff, S. Sanati, E. Christner, A. Zahn, M. Balzer, H. Bouquet, J. B. Mc-Manus, Y. González-Ramos, and M. Schneider. Airborne in situ vertical profiling of HDO / H216O in the subtropical troposphere during the MUSICA remote sensing validation campaign. *Atmospheric Measurement Techniques*, 8(5): 2037–2049, may 2015. ISSN 1867-8548. doi: 10.5194/amt-8-2037-2015. URL http://www.atmos-meas-tech.net/8/2037/2015/.

J. Galewsky, C. Rella, Z. Sharp, K. Samuels, and D. Ward. Surface measurements of upper tropospheric water vapor isotopic composition on the Chajnantor Plateau, Chile. *Geophysical Research Letters*, 38(17), sep 2011. ISSN 00948276. doi: 10.1029/2011GL048557. URL http://www.agu.org/pubs/crossref/2011/2011GL048557.shtml http://doi.wiley.com/10.1029/2011GL048557.

K. E. Samuels-Crow, J. Galewsky, Z. D. Sharp, and K. J. Dennis. Deuterium excess in subtropical free troposphere water vapor: Continuous measurements from the Chajnantor Plateau, northern Chile. *Geophysical Research Letters*, 41 (23):8652–8659, dec 2014. ISSN 00948276. doi: 10.1002/2014GL062302. URL http://doi.wiley.com/10.1002/2014GL062302.

---

## Referee Comment (RC2) · Anonymous Referee #3 · 18 Jan 2017

General Comments

High resolution in situ aircraft based isotopic (18O/16O and 2H/1H) measurements of atmospheric water vapor between 150 and 4500 m a.s.l. using a laser based optical analyzer are reported. The low abundance of water vapor in upper tropospheric air, the large range of concentrations and the low abundances of Oxygen-18 and Deuterium render such measurements difficult and referee 1 raises some points of concern (Teflon is mentioned wrt to memory effects, perhaps the filter used (Figure A2) should be described. The interpretation of the experimental data in the light of adequate meteorological information is complete and attention is paid to proper experimental procedures for calibration. Although the reader needs some patience to follow all details of the many series measured the paper does justice to the data and will be valuable for those embarking on acquiring and using similar data. An important step is made by explaining profiles as the result of mixing of air masses and not only local Raleigh type stable isotope fractionation. One question a reader may have is why not any other data from these flights were used. In other words, the isotope data and physical data are used together, without any other tracer data. It may well be the unique nature of water vapor and its isotopic composition that make it hard to find any other tracer that supports interpretation.

Below are my (mostly technical) comments.

The title should be "The stable isotopic composition of water vapour above Corsica during the HyMeX SOP1 campaign.. .."

"Stable water isotopes" I think this sloppy descriptor ought to be removed from the paper because it is wrong. We get throughout the paper statements like: Stable water isotopes, stable water isotope composition, stable water isotope profiles, the SWI composition of atmospheric water vapor (page2, line 18) and so on. Does SWI stand for vapor or liquid? Better is to write what it is, use isotopic composition (IC), for instance the IC of snow, or the IC of water vapour. In case one is concerned to wrongly generate the impression that Tritium measurements were involved, then use SIC. Or one can use HIC for Hydrogen isotopic composition and OIC . . . such logical abbreviations will not be forgotten for years to come, and do not sound as bad as some recent Twitter messages.

The delta values are defined (since at least 6 decades) as atomic ratios. Reading the introduction, a reader may think that delta values are based on molecular ratios. The laser analyses absorption features based on molecular properties. Which standards are used, and how. Why does WS9 have a large deuterium excess?

For non isotope colleagues (stable), perhaps explain why delta d is use and not delta 18O. The 18O/16O ratio is larger than the D/H ratio (which is twice the DHO/H2O ratio).. Is the systematic error in the d- excess due to a noisier Deuterium or 18O signal? On page 8, line 23 delta 18O is used and not delta D.

In the introduction it is emphasized that these are the first airborne spectroscopic stable water isotope measurements over the Mediterranean. The reader may well think, "what about the other seas and oceans, have they been left out? The coveted quantifier "first" can be used by narrowing down in space. But it helps no-one.

Page 2, line 8. To what does "these" pertain?

Page 2, line 11. I am sure that by far most delta 18O and delta D measurements have been made for hydrological purposes, e.g. precipitation network, ground water and aquifer studies, not palaeoclimate.

Page 3, line 31. "advection" Do the authors mean diffusion? What exactly is advection and why would that fractionate? Convection does not fractionate, neither does advection, I suspect.

Page 5, line 26. This sentence means that humicap (should it be Humicap?) provides slow accurate measurements. Is this true? These are small sensors that need calibration. How can they be accurate? I assume only if they have a stable response from before to after a measurement series.

Page 6, line1. "installation of a replacement pump" It is a bit unclear what has happened. Referred to is A2, where is written "not shown".

Page 6. What is q?

Page 6, line 31. What is the source of such very precise numbers, e.g. -78.68 per mil for deuterium.

Page 7, line 27. ..lower and higher.

Page 9, line 31. Replace "air" by water vapour" and replace "value" by "values".

Page 10, line 2. The deuterium excess is not measured directly, but I think derived from the D and 18O signals. Can you pinpoint which has most influence on the deteriorating precision and accuracy?

Page 10, line 6. The sentence starting with "Only few.." can be deleted.

Page 10, line 11. "remarkable" is perhaps not the correct description. Ehhalt was a very good experimental scientist. The reader has problems to get convinced. The red dots (Fig. 5 (a)) roughly fall in the measured range, that is all. Is a curve through the mean or median values not better? Also, in the same figure box, we do not see a zero gradient between 0 and 1500. A small decrease is visible.

Page 10, line 28. Perhaps replace "common" by "earlier" and insert "is based on".

Page 11, line 1. "is constant up to almost 1200 m a.s.l." OR "is almost constant up to about 1200 m a.s.l.).

Page 11, line 3. Here is suddenly written "major isotope species" This sentence needs to be corrected. "with their very depleted conditions" (sounds like people being robbed).

Page 12, lines 27-29. The argument is not convincing and referee #1 mentions the problem of the d-excess data. I do not know what to advice here.

Page 14, line 1. Please insert (Fig. 1) after "pattern #3".

Page 14, line 18. Which flight?

Page 16, line 20. Why flight 10?

Page 19. Summary and Conclusions. I am sorry, but the conclusions need to be partly rewritten. There are valuable findings which allow some conclusions and these conclusions should not be cluttered up with much less relevant information.

Line 3. add "campaign".

Line 3. It may be the first such data for Corsica, but it will not make Napoleon come back. Safe and appropriate to write is that it is if not the first, one of the first extensive airborne datasets in the framework of a well documented measurement campaign. Later is written, that your finding is confirmed by Dyroff, who actually published BEFORE you did. Please change.

Line 19. It is a bit hard on Claude Taylor, who after all pioneered similar isotope measurements over 4 decades ago. The statement does not make the paper or experimental work more valuable.

Line 26. "non-linearities in the delta scale" This needs an explanation, are there more than one type on non-linearities. Perhaps deal with this issue earlier in the paper, explaining the delta values and ratios.

References. perhaps useful: Paired stable isotopologues in precipitation and vapor: A case study of the amount effect within western tropical Pacific storms Conroy, JL et al. JOURNAL OF GEOPHYSICAL RESEARCH-ATMOSPHERES. DOI: 10.1002/2015JD023844

---

## Author Comment (AC1) · 7 Mar 2017

Reply to reviewer comments for manuscript acp-2016-728

**The stable isotope composition of water vapour above Corsica during the HyMeX SOP1: insight into vertical mixing processes from lower-tropospheric survey flights**

Harald Sodemann, Franziska Aemisegger, Stephan Pfahl, Mark Bitter, Ulrich Corsmeier, Thomas Feuerle, Pascal Graf, Rolf Hankers, Gregor Hsiao, Helmut Schulz, Andreas Wieser, and Heini Wernli

**Reply to reviewer #1:**

*We thank the reviewer for his/her detailed and constructive comments on our measurement and calibration approach, which helped improving the clarity of our manuscript. Our replies to the individual comments are in italics.*

General comments

[...] I recommend publication of this innovative, very well-written paper after my following major two concerns regarding measurement uncertainty have been addressed:

1.In my opinion, a central assumption of this publication regarding the measurements of deuterium excess is stability of the deuterium excess-humidity response of the instrument. The applied deuterium excess-humidity response correction was determined after the campaign. The fact that this response is individual for each Picarro Isotope Analyzer may indicate that the response is related to the alignment of optical components or other parameters that could change with time or in consequence of mechanical stress. Because the magnitude of the applied humidity correction for deuterium excess (Fig.A3c) is of the same order of magnitude as the observed range of observations of deuterium excess (Fig.6b) this may introduce significant systematic uncertainty to the observations. The authors mention additional calibrations of the SWI-humidity response during the campaign. Could you add them to Fig.A3 to demonstrate stability of the SWI-humidity response correction?

*This is a valid point, and yes the additional calibrations of the isotope-humidity response do demonstrate the stability of this correction function. The isotope-humidity response was checked for low (~4000ppmv) and high humidity levels (~20'000ppmv) during the campaign in the field on 26.9.2012 and over a more complete range after the campaign on 25.10.2012 in the laboratory. A further extended humidity range was covered with a refined setup in 2015 in the laboratory which is the basis of the the isotope-humidity response used for correcting our data. Fig. A3 already contains some of these data points, but our description was incomplete. Both calibration sequences on 26.9.2012 and 25.10.2012 suffered from remaining background humidity in the dried ambient air used in the calibration, which limit the range of useful data points down to about 4400ppmv. We have revised Fig. A3 to include the data points from the earlier calibration run, and updated the description in Appendix A3. We do not show the standard deviation of the preliminary calibrations of the isotope humidity response to avoid unnecessary cluttering of the figure.*

2. Uncertainty estimates for the SWI measurements are based on the standard deviation of calibration standard measurements in the lab (Fig. A4). However, this estimate does not account for (a) memory effects of the inlet tubing and other effects responsible for the observed hysteresis of SWI measurements, (b) potentially increased instrumental drift in consequence of mechanical stress during the flights, and (c) systematic uncertainty of the applied SWI-humidity response correction. Could you quantify these three uncertainty terms e.g. for humidity levels of 1000, 5000, and 10000 ppmv:

(a) The observed hysteresis of SWI measurements should be consistent with the stated measurement uncertainty. Could you quantify the observed mean hysteresis of the SWI measurements for different humidity levels?
In my opinion, not only changes in cavity pressure but also memory effects of inlet tubing and analyzer are reasonable explanations for the observed strong hysteresis of the SWI measurements. The 123cm of inlet tubing closest to the isotope analyzer were not flushed by the TF2 flushing pump. Considering the small air stream of 0.1 l min$^{-1}$ through this part of the tubing, the relatively strong adsorption of water on PTFE

surfaces, and encountered strong gradients of specific humidity memory effects may play an important role. This seems to be supported by the differences between SWI measurements during ascent and descent, which appear to be especially pronounced during dry conditions and smaller during moist conditions. Compare e.g. Fig. 9g: dry conditions above 1700m a.s.l. are related to a strong hysteresis of $\delta D$, wet conditions below 1700 m a.s.l. are related to a small hysteresis of $\delta D$.

*We agree that memory effects have not been sufficiently taken into account as an explanation of the hysteresis observed in Fig. 9. Wall effects of the PTFE tubing could become more relevant at lower humidity. We added such a statement to Sec. 2.5 and Sec. A4, and modified p.15,L.33-34, p.21,L.18-20 (locations in the original manuscript given by reviewer #1).*

With respect to memory effects of the measurement system the authors state time constants of 2.4s, 1.7s, and 1.3s for $\delta D$, $\delta^{18}O$, and H16O, which are within the applied averaging time of 15 s. These time constants were determined in Aemisegger [2013]. As I understand it, Aemisegger [2013] derived the time constants by measuring the instrument response to changes of specific humidity of 1000ppmv between 12000 and 22000ppmv. I assume that these time constants may be substantially larger for drier conditions and larger humidity steps e.g. from 5000 to 500 ppmv.

*In Aemisegger (2013), two switching experiments are described, in her Sec. 2.7 an experiment with a Picarro instrument L1115-i with humidity changes of 1000ppmv between 12000 and 22000ppmv, and in her Sec. 7.2.5 one with the same inlet system as used in the aircraft measurements, and larger switches, approximately between 3000 and 12000ppmv. We agree that the time constant could be larger at drier conditions, but lack laboratory measurements of such humidity changes. A sentence describing this aspect and mentioning potential implications has been added to section 2.5.*

(b) During the campaign SWI calibrations were performed before and after the flights. However, most of the calibrations before the flights had to be discarded because stabilization of the instrument was not completed. Could you estimate typical drift of the isotope analyzer during the flights based on the few days with successful calibrations before and after a flight?

*We had in total 5 days where calibration was performed before and after flights on the same day. Six bracketing calibrations were done at approximately 20'000ppmv, and only two at 4000ppmv. Assuming linear drift between the calibrations, the overall drift was about 0.2 permil $h^{-1}$ for $\delta^{18}O$, 0.5 permil $h^{-1}$ for $\delta D$, and independent of the stable isotope concentration. Drift was about 3x stronger at 4000ppmv, but this number is based on only two calibrations from the same day. A discussion to this end was added to Sec. A4.*

(c) Do the errorbars in Fig. A3 state the total uncertainty of the SWI-humidity response calibration? Could you underpin stability of the SWI-humidity response calibration by measurements during the campaign?

*The errorbars represent the measurement uncertainty of each data point. The 95% confidence bounds of the response function from the fitting procedure is now included as thin lines in Fig. A3c. We can thus quantify the additional error contribution to the d-excess at 9000ppmv, 5000ppmv and 2000ppmv as 0.5‰, 5‰ and 20‰. Measurements from during the campaign are now incuded in the figure as colored dots without error bars, as stated in the reply to General comment #1.*

*We have included a paragraph discussing the additional uncertainty from the 3 aspects pointed out by the reviewer in Sec. A4.*

**Specific comments**

p.2,L.8: In my opinion, stating the definition of the $\delta$-notation or citing e.g. [Coplen, 2011] would be helpful.

*We added a citation of Coplen (2011) to the introduction.*

p.6,L.1: "...changed after the installation of a replacement pump..." Slope and offset between CRDS humidity measurements and HCLY measurements were stable between flights, but changed after the

installation of a replacement pump. Does this change indicate memory effects of the inlet tubing, whereby the impact of these effects changed in magnitude in consequence of an adjusted air flow?

*Possibly the previous pump was already performing less than ideal before it had to be replaced because of failure. The change in the calibration parameters is shown below for reference. A slight change appeared for the slope parameter, which could be related to a change in the flow rate, but the time offset was nearly constant which does not suggest a strong effect of the replaced pump on memory effects (Figure R1). There is more speculation than evidence in this because only 5 flights were performed with the replacement pump, but we note it in the manuscript to as the pump change is potentially relevant for our data set. We expand the statement in Appendix A2 as follows in the revised manuscript:*

*"The offset of this linear fit was mostly stable between flights, whereas the slope changed slightly from 1.0 to about 1.1 after the installation of a replacement pump since the original pump was broken between flights #17-27. This could affect the pressure in the inlet system, resulting in a different regulatory behaviour and pressure hysteresis of the mesurement setup, but does not indicate a change in flow rate (not shown). This may suggest that calibration should ideally cover the entire inlet system, including the by-pass pump."*

[Figure]

***Figure R1:*** *Slope, offset and correlation coefficient from a linear regression between $q_{HCLY}$ and $q_{CRDS}$ for the flights of the HyMeX KIT campaign.*

p.6,L.10-28: See my first point in the general comments.

*See reply to general comment 1.*

p.6,L.30-p.7,L.10.: SWI measurements with the Picarro isotope analyzer are based on linear two-point calibrations at $\delta D=-78.68$ and $-166.74$. Aemisegger et al. [2012] show significant non-linearity of this correction for an older version (L1115- i) of the Picarro isotope analyzer. As many of the observations with high d-excess are related to $\delta D$ ratios smaller than $-166.74$ (Fig.4) it might be of interest to quantify respective additional measurement uncertainty at small isotope ratios.

*Indeed, the standards did not span the entire range of depletions encountered during flight conditions. In Aemisegger et al., (2012), it is shown that the uncertainties of the isotope standard measurements from two laser spectroscopic systems obtained from error propagation after calibration depend on the δ value of the chosen calibration standard (their Fig. 3). The additional error for data points into the range outside of the*

*calibration bracket is however comparatively small, and dominated by the error components during measurement of the standards with a calibration device in the field.*

p.7.,L.15-L.18: The required measurement accuracies of H16O, δD, and especially of deuterium excess are of different order of magnitude. A drift of the CRDS H16O calibration by 10 would be almost undetectable if comparing the CRDS and HCLY humidity measurements. However, changing δD and δ18O by 10 would shift the deuterium excess even for 70 . I therefore think that a small observed drift of the CRDS H16O calibration does not necessarily confirm a small drift of the SWI calibration.

*This is a valid point. The main point of our argument is that to first order, the results are not showing an obvious influence of the flight conditions on the spectroscopic measurements. We weakened the respective statement in the revised manuscript accordingly.*

p.7,L.22-24: Dyroff et al. [2015] used a different measurement principle. For this reason please skip this sentence.

*We removed the sentence from the revised manuscript.*

p.7,L.26-28: "From laboratory experiments with the inlet system..." Please add a description of this experiment to the Appendix. See my point 2a in the general remarks.

*We have added a description of this experiment now in Sec. 2.5, and formulated that increased time constants at low humidity and low air pressure can contribute to increased memory during upward profiles into drier air layers.*

p.9,L.33: Could you explain in more detail what you mean by non-linearity of the delta scale.

*The d-excess is usually considered as a measure of kinetic fractionation effects, but can also change under pure equilibrium conditions due to the temperature effect on the ratio between the fractionation factors of δ18O and δD, and the nonlinearity of the delta-scale. The latter becomes apparent under strongly depleted conditions. A simple example to illustrated this aspect is that in the limiting case of a totally depleted sample (meaning an isotope ratio R of 0 for both δ18O and δD) one would obtain delta values for both isotopes of -1000‰ and thus a deuterium excess of +7000‰, in absence of any kinetic fractionation. We have included a sentence in the Introduction that gives an indication of this aspect:*

*"Note however that at highly depleted conditions with respect to HDO and $H_2^{18}O$, as encountered at higher tropospheric levels, the d-excess can take high values because of the non-linearity of the scale it is defined on, even in absence of kinetic fractionation (the d-excess becomes 7000‰ in the limiting case of -1000‰ for both HDO and $H_2^{18}O$). Therefore, careful interpretation of this parameter is required. "*

p.9,L.31-p.10,L.1: "The high d-excess ... is therefore not an indication of insufficient data quality, but a real feature ... in the atmosphere": I agree that the high d-excess encountered is not an indication of insufficient data quality, but I don't think that agreement with the observations of Galewsky et al. [2011] and Samuels-Crow et al. [2014] alone justifies the conclusion that the SWI composition presented here is a real feature. I would rather treat this agreement as further evidence for reliable observations. Please rephrase this sentence.

*Correct, we rephrased that sentence.*

p.12,L.6: "a very dry and depleted free tropospheric air mass": The free tropospheric end member you assume for the green mixing curve is drier but less depleted than the free tropospheric end member corresponding to the orange curve. Can the free tropospheric end member corresponding to the green mixing curve be explained by condensation alone or do the specific humidity of 0.5 g kg$^{-1}$ and the δD of −220‰ imply earlier mixing?

*The depleted end-member of the green mixing line can not be explained by a Rayleigh model for the surface conditions encountered within our data set (specific humidity at the surface >3 g kg⁻¹ and δD >-140‰). We include this important point in the revised manuscript.*

p.14,L.34: As mentioned above I would appreciate a quantification of the memory effects regarding uncertainty of the SWI measurements.

*This comment has been taken into account in General remark 2a.*

p.15,L.33-34: "Reproducibility was good ... except for the dry intermediate layer". To me this seems like further evidence for a humidity-dependent and not a cavity pressure-dependent hysteresis of the SWI measurements.

*We address this topic in General comment 2a above.*

p.20,A1: Was temperature in the non-pressurized cabin stabilized? Did you heat the tubing to avoid condensation?

*No, temperature in the non-pressurised cabin was not stabilised, and the tubing was not heated. However, temperatures inside the cabin were generally higher than ambient air temperature outside due to insolation. It is therefore unlikely that condensation in the tubing took place. We added a statement to this end to section A1.*

p.21,L.3-5: I wouldn't use the standard deviation of calibration standard measurements as measure for the total uncertainty of the observations during the campaign as this value doesn't consider the effects causing the observed hysteresis of SWI measurements, potentially increased instrumental drift during the flights in consequence of mechanical stress, and systematic uncertainty of the humidity response correction. See my second point in the general remarks.

*We now take these additional factors into account in this paragraph and the overall discussion of the results as detailed in the reply to the general remarks.*

p.21,L.18-20: "Possible causes of this hysteresis... due to observed changes in the cavity pressure": What about memory effects causing the hysteresis?

*We have added a discussion of the memory effect as a cause for hysteresis as mentioned above.*

p.21,L.32-p.22,L.4: The authors refer to SWI calibrations bracketing each flight ensuring stability of the SWI measurements. This is in contrast to p.7,L.6-7: "...with very few exceptions only the calibration runs after the daily flight operations were used...". Maybe you could add a figure demonstrating the stability of SWI measurements based on the few days with successful calibrations before and after the flights. Otherwise I would skip the sentence p.22,L.2-4 "...calibration during flight would not improve data quality substantially in our case..."

*Generally, the uncertainty of our data is not dominated by the drift between the calibration runs but rather by (1) the humidity correction, (2) the signal/noise ratio (precision) and (3) by the response time especially at low humidities. During the 4 days when bracketing calibrations were available, the drift was typically 0.2 permil h⁻¹ for δ¹⁸O, and 0.5 permil h⁻¹ for δD, as mentioned above. If bracketing calibrations were performed during and after flight, these relatively small drifts could be corrected by calibration on the ground. However, conditions during flight are not as stable as on ground and such calibration runs may suffer from deterioration of measurement precision. Also, it is important to keep in mind that inflight calibration implies loss of valuable insitu data and has questionable representativity since the conditions (mechanical stress, slight temperature fluctuations, inlet pressure) during flight are constantly changing. Longer flights than in our campaign might however argue for in-flight calibration, and we moderated the statement on p.22 slightly.*

Fig. 2a: What is the meaning of the red and black lines.

*Red line is linear least-squares fit to the $q_{CRDS}$ and $q_{HCLY}$ data, black line is a 1:1 relation. This has been added to the caption of Fig. 2 in the revised manuscript.*

Fig2,Fig3,Fig6,Fig9,Fig10,Fig11,Fig12,FigA3,FigA4,FigA5: I would appreciate consistent use either only of ppmv or only of g kg$^{-1}$ for stating humidity in the different figures and in the text.

*We use units of g kg$^{-1}$ for humidity on all plots. In addition we show a ppmv scale on Fig. 6, A3, A4 and A5 because it is here particularly useful for an easier comparison with other published studies that use units of ppmv and that are directly reported by the used instrumentation. The g kg$^{-1}$ scale went missing in Fig. A4 but will be added in the revision. We added a note to the caption of Fig. 6.*

Fig6b: Could you add systematic uncertainty of the deuterium excess measurements (resulting e.g. from uncertainty of the SWI-humidity response correction) to this figure?

*Error bars for the uncertainty resulting from the isotope response correction have been added to Fig. 6b.*

Fig.A3: Do the error bars show statistical uncertainty from measurement noise? What is the meaning of the red crosses?

*The red data points were obtained from a calibration sequence during the HyMeX field campaign in 2012. The crossed out points are affected by incomplete removal of ambient moisture. Vertical bars show the standard deviation of each calibration point with measurements lasting for approximately 6–10min. Statements to this end clarifying this have been added to the caption of Fig. A3.*

**Technical corrections**

p.7.,L.30: CRDS

*done*

p.8,L.4: "1.17": Table 3 states an uncertainty of 1.18 for d-excess.

*done*

p.11,L.31: "relatively moist": Do you mean relatively dry?

*yes, done*

p.11,L.35: "magenta line": The respective line is orange.

*done*

p.21,L.4: "Figure A5": Figure A4?

*done*

Fig 9: "black or solid ... grey or dashed lines": There are no dashed lines visible in this figure.

*Changed to "Only downward profiles are shown as they are less affected by memory than upward profiles."*

Fig. 10: "red stippled": not shown

*Deleted*

Fig. A5: "red": magenta; "blue": green

**Reply to Reviewer #2**

*We thank the reviewer for his/her detailed and constructive comments, which helped improving the clarity of our manuscript. Our replies to the individual comments are in italics.*

General Comments

High resolution in situ aircraft based isotopic (18O/16O and 2H/1H) measurements of atmospheric water vapor between 150 and 4500 m a.s.l. using a laser based optical analyzer are reported. The low abundance of water vapor in upper tropospheric air, the large range of concentrations and the low abundances of Oxygen-18 and Deuterium render such measurements difficult and referee 1 raises some points of concern (Teflon is mentioned wrt to memory effects, perhaps the filter used (Figure A2) should be described. The interpretation of the experimental data in the light of adequate meteorological information is complete and attention is paid to proper experimental procedures for calibration. Although the reader needs some patience to follow all details of the many series measured the paper does justice to the data and will be valuable for those embarking on acquiring and using similar data. An important step is made by explaining profiles as the result of mixing of air masses and not only local Raleigh type stable isotope fractionation. One question a reader may have is why not any other data from these flights were used. In other words, the isotope data and physical data are used together, without any other tracer data. It may well be the unique nature of water vapor and its isotopic composition that make it hard to find any other tracer that supports interpretation.

*No other tracer data was available on these flight. It may be interesting to complement the analysis by CO, CH4 or other trace gases, but the history of phase changes and moisture origin is only reported by the water isotope composition. We mention this in the introduction of the revised manuscript and in Sec. 2. We also add a description of the Teflon filter we used to section A1.*

Below are my (mostly technical) comments.

The title should be "The stable isotopic composition of water vapour above Corsica during the HyMeX SOP1 campaign.. .."

*The title has been modified as suggested.*

"Stable water isotopes" I think this sloppy descriptor ought to be removed from the paper because it is wrong. We get throughout the paper statements like: Stable water isotopes, stable water isotope composition, stable water isotope profiles, the SWI composition of atmospheric water vapor (page2, line 18) and so on. Does SWI stand for vapor or liquid? Better is to write what it is, use isotopic composition (IC), for instance the IC of snow, or the IC of water vapour. In case one is concerned to wrongly generate the impression that Tritium measurements were involved, then use SIC. Or one can use HIC for Hydrogen isotopic composition and OIC . . . such logical abbreviations will not be forgotten for years to come, and do not sound as bad as some recent Twitter messages.

*The term "Stable water isotopes (SWI)" is a simplified descriptor that is intended to make the complex topic of stable isotopes more accessible to a wider audience. We agree, however, that for this study the advantage from using this term is limited. We therefore rephrased all occurrences of SWI by more specific terms.*

The delta values are defined (since at least 6 decades) as atomic ratios. Reading the introduction, a reader may think that delta values are based on molecular ratios. The laser analyses absorption features based on molecular properties. Which standards are used, and how.

*We have included a statement explaining this aspect to the introduction: "It should be noted that the isotope ratios determined by laser spectroscopy are molecular isotope ratios. In practical cases it can be shown that the difference between the molecular ratios and the more commonly used atomic ratios is much smaller than the measurement precision (Kerstel, 2004)."*

*The atomic ratios [D]/[H] and [18O][16O] are thus assumed to be equivalent to the molecular ratios [1H16O2H]/(2\*[1H16O1H]) and [1H18O2H]/[1H16O1H] from which our molecular delta values are derived. The factor 2 in the molecular R2H comes from the two possible positions of the hydrogen atom in the H2O molecule.*

Why does WS9 have a large deuterium excess?

*It is an artificially produced mixture of waters.*

For non isotope colleagues (stable), perhaps explain why delta d is use and not delta 18O. The 18O/16O ratio is larger than the D/H ratio (which is twice the DHO/H2O ratio). Is the systematic error in the d-excess due to a noisier Deuterium or $^{18}$O signal? On page 8, line 23 delta 18O is used and not delta D.

*We generally show d-excess and $\delta D$ because fractionation is stronger for $\delta D$, because $\delta^{18}O$ is more strongly affected by non-equilibrium fractionation than $\delta D$. There is no systematic error arising from the noisiness of a signal. The impact of the $\delta^{18}O\text{-}H_2O$ correction on the d-excess is stronger than the one coming from $\delta D$, but both $\delta D$ and $\delta^{18}O$ play a role. We now use $\delta D$ also for the characterisation of the river runoff, and make a statement explaining why we focus on $\delta D$ and d-excess in Sec 2.5:*

*"We focus discussion of our own measurements mostly on the complementary parameters $\delta D$ and d-excess (the d-excess measures how $\delta^{18}O$ deviates from the behaviour expected at equilibrium conditions)."*

In the introduction it is emphasized that these are the first airborne spectroscopic stable water isotope measurements over the Mediterranean. The reader may well think, "what about the other seas and oceans, have they been left out? The coveted quantifier "first" can be used by narrowing down in space. But it helps no-one.

*We agree with the reviewer and the word first is not required here. Our general intention is to allow the results to speak for themselves within a scientific context that brings forward their value and meaning, and we are grateful for this moderating comment. The respective sentences in the abstract, introduction and conclusions have been rephrased accordingly.*

Page 2, line 8. To what does "these" pertain?

*We added a missing sentence mentioning the fractionation during phase changes*

Page 2, line 11. I am sure that by far most delta 18O and delta D measurements have been made for hydrological purposes, e.g. precipitation network, ground water and aquifer studies, not palaeoclimate.

*It may be difficult to defend this statement, but at the same time it should be noted that many precipitation samples have been measured in support of paleoclimate interpretation, and that one single ice core can require thousands of measurements to complete. The claim is however not crucial to this manuscript, and we rephrased the sentence to avoid misunderstandings:*

*"In the past, some of the most prominent use of stable isotopes has been in a palaeoclimate context to infer past temperatures and moisture sources from natural archives (e.g., Jouzel et al., 1997), for ground water studies (e.g., Sonntag et al., 1983), and in studies investigating the stratospheric water budget (e.g., Webster and Heymsfield, 2003). "*

Page 3, line 31. "advection" Do the authors mean diffusion? What exactly is advection and why would that fractionate? Convection does not fractionate, neither does advection, I suspect.

*This was meant to be "diffusion" and has been corrected.*

Page 5, line 26. This sentence means that humicap (should it be Humicap?) provides slow accurate measurements. Is this true? These are small sensors that need calibration. How can they be accurate? I assume only if they have a stable response from before to after a measurement series.

*The Humicap is typically calibrated once for an entire campaign, as the calibration curve remains very stable on a day-to-day basis and more frequent calibration is not practical (calibration takes one half to one day). Calibration was however done after low flights when salt deposits required cleaning of the sensors. Calibration was performed over saturated saline solutions. We added a sentence to this section: "The Humicap was calibrated repeatedly during the campaign and showed very stable calibration curves."*

Page 6, line1. "installation of a replacement pump" It is a bit unclear what has happened. Referred to is A2, where is written "not shown".

*We explain this now in more detail in Sec. A2:*

*"The original flush pump failed during flight 17, and was replaced with the same type of pump before flight 28. The slope of the humidity calibration changed from 1.0 to about 1.1 after the installation of the replacement pump which could indicate a change in flow rate (not shown)."*

Page 6. What is q?

*This was corrected to be $w_{raw}$ in Eqs 1 and 2 which is the uncalibrated volume mixing ratio of water vapour from the CRDS in units of ppmv.*

Page 6, line 31. What is the source of such very precise numbers, e.g. -78.68 per mil for deuterium.

*The working standards were obtained from repeated isotope-ratio mass spectrometry measurements. We have reduced the standard values to one decimal place.*

Page 7, line 27. ..lower and higher.

*Corrected.*

Page 9, line 31. Replace "air" by water vapour" and replace "value" by "values".

*Corrected.*

Page 10, line 2. The deuterium excess is not measured directly, but I think derived from the D and 18O signals. Can you pinpoint which has most influence on the deteriorating precision and accuracy?

*This is now mentioned in the introduction, see reply to comment "For non-isotope colleagues...". We also extended the discussion in Sec. A4 to include the effect of temporal averaging on the precision. The precision of $d^{18}O$ is approximately 5-6 times better than the precision of dD (see standard deviations at different averaging times). So $d^{18}O$ has the strongest influence on the precision of the d-excess.*

Page 10, line 6. The sentence starting with "Only few.." can be deleted.

*Sentence deleted.*

Page 10, line 11. "remarkable" is perhaps not the correct description. Ehhalt was a very good experimental scientist. The reader has problems to get convinced. The red dots (Fig. 5 (a)) roughly fall in the measured range, that is all. Is a curve through the mean or median values not better? Also, in the same figure box, we do not see a zero gradient between 0 and 1500. A small decrease is visible.

*We are not sure why the reviewer mentions that Ehhalt was a very good experimental scientist, it was certainly not our intention to put this into question. The word "remarkable" was chosen to express our astonishment about the general correspondence of how the range of δD measurements changes with altitude when comparing two separate places and time periods. We rephrased to avoid misunderstandings. The negative gradient is already described in Sec. 3 when discussion Fig. 4.*

Page 10, line 28. Perhaps replace "common" by "earlier" and insert "is based on".

*Rephrased as suggested.*

Page 11, line 1. "is constant up to almost 1200 m a.s.l." OR "is almost constant up to about 1200 m a.s.l.).

*Rephrased according to the second option.*

Page 11, line 3. Here is suddenly written "major isotope species" This sentence needs to be corrected. "with their very depleted conditions" (sounds like people being robbed).

*We rephrased and removed the confusing use of the word "depleted".*

Page 12, lines 27-29. The argument is not convincing and referee #1 mentions the problem of the d-excess data. I do not know what to advice here.

*Reviewer #1 does not state that the d-excess values are/were and indication of insufficient data quality, but rather highlights potential sources of uncertainty. We added a statement referring to the discussion of high d-excess in Sec. 3 which has now been expanded to include a brief discussion of the non-linearities of the d-excess.*

Page 14, line 1. Please insert (Fig. 1) after "pattern #3".

*Inserted as suggested.*

Page 14, line 18. Which flight?

*Inserted information on flight 09.*

Page 16, line 20. Why flight 10?

*We added a statement explaining that this was the flight with the most pronounced isotope gradient and temperature inversion.*

Page 19. Summary and Conclusions. I am sorry, but the conclusions need to be partly rewritten. There are valuable findings which allow some conclusions and these conclusions should not be cluttered up with much less relevant information.

*We re-wrote much of the summary and conclusions to focus on the most relevant findings, see below.*

Line 3. add "campaign".

*done*

Line 3. It may be the first such data for Corsica, but it will not make Napoleon come back. Safe and appropriate to write is that it is if not the first, one of the first extensive airborne datasets in the framework of a well documented measurement campaign. Later is written, that your finding is confirmed by Dyroff, who actually published BE- FORE you did. Please change.

*We rephrased these sentences to highlight the main improvement compared to earlier studies (high-resolution measurements), and corrected the use of the word "confirmed".*

Line 19. It is a bit hard on Claude Taylor, who after all pioneered similar isotope measurements over 4 decades ago. The statement does not make the paper or experimental work more valuable.

*It was by no means the intention here to criticise or diminish the impressive, pioneering work of Claude Taylor, but to point out the finding that airborne d-excess measurements contain potentially useful information. We have rephrased to emphasise that aspect only.*

Line 26. "non-linearities in the delta scale" This needs an explanation, are there more than one type on non-linearities. Perhaps deal with this issue earlier in the paper, explaining the delta values and ratios.

*We now address the question of non-linearities in the discussion of high d-excess values at low humidity conditions, see reply to Reviewer 1.*

References. perhaps useful: Paired stable isotopologues in precipitation and vapor: A case study of the amount effect within western tropical Pacific storms Conroy, JL et al. JOURNAL OF GEOPHYSICAL RESEARCH-ATMOSPHERES. DOI: 10.1002/2015JD023844

*We consider including this reference in the revised manuscript.*

---

## Author Comment (AC2) · 7 Mar 2017

We thank the reviewer for his/her detailed and constructive comments, which helped improving the clarity of our manuscript. The replies to the reviewer's comments are included in the pdf document uploaded above for reviewer #1.